# The ubiquitin ligase Cul5 regulates CD4+ T cell fate choice and allergic inflammation

Binod Kumar[1,2,12], Natania S. Field[1,3,12], Dale D. Kim [1], Asif A. Dar[1], Yanqun Chen[1,4], Aishwarya Suresh [1,5], Christopher F. Pastore [6], Li-Yin Hung[6], Nadia Porter[1], Keisuke Sawada[1,7], Palak Shah[1,8], Omar Elbulok[1,9], Emily K. Moser[1,10], De'Broski R. Herbert [6] & Paula M. Oliver [1,11✉]

Antigen encounter directs CD4+ T cells to differentiate into T helper or regulatory cells. This process focuses the immune response on the invading pathogen and limits tissue damage. Mechanisms that govern T helper cell versus T regulatory cell fate remain poorly understood. Here, we show that the E3 ubiquitin ligase Cul5 determines fate selection in CD4+ T cells by regulating IL-4 receptor signaling. Mice lacking Cul5 in T cells develop Th2 and Th9 inflammation and show pathophysiological features of atopic asthma. Following T cell activation, Cul5 forms a complex with CIS and pJak1. Cul5 deletion reduces ubiquitination and subsequent degradation of pJak1, leading to an increase in pJak1 and pSTAT6 levels and reducing the threshold of IL-4 receptor signaling. As a consequence, Cul5 deficient CD4+ T cells deviate from Treg to Th9 differentiation in low IL-4 conditions. These data support the notion that Cul5 promotes a tolerogenic T cell fate choice and reduces susceptibility to allergic asthma.

[1] Division of Protective Immunity, Department of Pathology and Laboratory Medicine, The Children's Hospital of Philadelphia, Philadelphia, PA, USA. [2] Department of Microbiology, Perelman School of Medicine, University of Pennsylvania, Philadelphia, PA 19104, USA. [3] Department of Medicine, Northwestern University, Chicago, IL 60611, USA. [4] Wuxi Apptec, 4751 League Island Blvd, Philadelphia, PA 191124, USA. [5] Drexel University College of Medicine, 2900 W Queen Ln, Philadelphia, PA 19129, USA. [6] Department of Pathobiology, University of Pennsylvania School of Veterinary Medicine, Philadelphia, PA 19140, USA. [7] University of Cincinnati, College of Medicine, Cincinnati, OH 45267, USA. [8] Harvard University, Boston, MA 12138, USA. [9] College of Arts and Sciences, University of Pennsylvania, Philadelphia, PA, USA. [10] Division of Pulmonary, Critical Care, and Sleep Medicine, Department of Medicine, University of Florida, Gainesville, FL 32610, USA. [11] Department of Pathology and Laboratory Medicine, University of Pennsylvania, Philadelphia, PA, USA. [12] These authors contributed equally: Binod Kumar, Natania S. Field. ✉email: paulao@pennmedicine.upenn.edu

CD4[+] T helper (Th) cells have a critical function in mounting an effective immune response against wide variety of pathogens[1]. They help to clear pathogens by secreting cytokines, activating innate immune cells, promoting cytotoxic CD8 T cell function and helping B cells to mount a humoral response[2,3]. During antigen exposure, CD4[+] T cells can also become T regulatory (Treg) cells and suppress immune responses to innocuous antigens or dampen inflammation once the pathogen has been cleared, thus preventing autoimmune diseases[4]. The unique ability of CD4[+] T cells to perform both pro- and anti-inflammatory functions is due to their ability to differentiate into various subtypes of Th and Treg cells. This differentiation depends on the cytokines present during T cell receptor (TCR) stimulation[1,5]. Cytokines, upon binding to their receptors, activate Janus kinase (Jak) and Signal Transducer Activator of Transcription (STAT) signaling pathways[6,7]. Once activated, phosphorylated STAT (pSTAT) proteins induce expression of transcription factors that establish Th cell lineages[7]. While the signaling cascades initiated by cytokines are critical for CD4[+] T cell differentiation, limiting responses to low levels of cytokine helps to focus T cell differentiation and prevent autoimmune and allergic disease[8].

Suppressor of Cytokine Signaling (SOCS) proteins are potent inhibitors of cytokine signaling. There are 8 members in this family (SOCS1-7 and Cytokine Inducible SH2 protein (CIS)), that have a C-terminal SOCS box motif and central SH2 domain[9–11]. Studies have shown that SOCS proteins regulate T cell fate by limiting cytokine signaling[12,13]. For instance, SOCS1 negatively regulates Th1 differentiation, and mice lacking SOCS1 die of inflammatory disease soon after weaning[14,15]. SOCS3 limits IL-23 signaling and Th17 differentiation[16], while SOCS2 restricts Th2 differentiation[17–19]. Furthermore, CIS has been shown to limit both Th2 and Th9 differentiation and lung inflammation during experimental asthma[20].

While these studies highlight the importance of SOCS proteins in limiting the cytokine signaling, how they function mechanistically remains poorly understood. Since cytokine signaling induces the phosphorylation of receptors and signal transducers[13], one way to limit cytokine signaling is to inhibit kinase activity. Indeed, SOCS1 and SOCS3 proteins have a kinase inhibitory region (KIR) and inhibit the kinase activity of Jak proteins[13,21,22]. However, other SOCS proteins lack this domain. All SOCS proteins have an SH2 domain and a SOCS box domain[12]. The SH2 domain allows them to bind with phosphorylated receptors to block recruitment of signaling intermediates[13]. The SOCS box domain allows them to interact with Cullin 5 (Cul5) and thus participate in a Cullin Ring Ligase 5 (CRL5) E3 ubiquitin ligase complex[23]. However, SOCS box proteins bind Cul5 with widely varying affinities. Thus, each of the SOCS box protein may have different methods for inhibiting cytokine signaling, and the extent to which they work in a complex with Cul5 remains unclear.

Cul5 is a scaffold protein that nucleates the CRL5 complex[24]. As with other Cullins, E3 ubiquitin ligase activity is activated by neddylation, a post-translational modification in which the ubiquitin-like protein Nedd8 is covalently attached to Cul5[25]. The CRL5 complex consists of, among other proteins, a RING-box (Rbx) protein that recruits the E2 ubiquitin conjugating enzyme, and a substrate receptor that recruit substrates[24,26,27]. SOCS proteins are thought to function as substrate receptors for the CRL5 complex[23,25], bringing the substrate and E2 in close proximity to allow substrate ubiquitination. Given the central function of SOCS proteins in regulating cytokine signaling, and their potential to act as substrate receptors for CRL5, we posited that Cul5 might be important in CD4[+] T cell differentiation.

Here, we show that Cul5 regulates CD4[+] T cell fate decisions. Supporting this mice lacking Cul5 in T cells (Cul5[fl/fl]CD4-Cre) exhibit a very mild Th2 inflammation that worsens with age. Following allergen challenge, Cul5[fl/fl]CD4-Cre mice show elevated levels of Th9 cells, an increase in eosinophilia, goblet cell hyperplasia, and airways resistance. In activated CD4[+] T cells, Cul5 associates with CIS to form a CRL5 complex. As has been shown for CIS-deficient cells, T cells lacking Cul5 are more likely to differentiate into Th2 and Th9 cells. In these cultures, Cul5-deficient T cells show increased expression of Th9-associated genes and evidence of increased STAT6 signaling. Following T cell activation, CIS recruits pJak1 to Cul5, resulting in decreased pJak1 levels and reduced IL-4R signaling. Limiting IL-4R signaling alters fate selection in CD4[+] T cells. Supporting this, Cul5-deficient cells preferentially develop into proinflammatory Th9 cells instead of tolerogenic Treg cells. These data support that Cul5 regulates fate choice in CD4[+] T cells and reduces susceptibility to allergic inflammation.

## Results

**Cul5 expression and activation is increased following CD4[+] T cell stimulation.** CRL5 ligase activity is turned on when the ubiquitin-like protein Nedd8 is covalently attached to Cul5 in a process called neddylation[27,28]. Using di-glycine remnant profiling, we previously found that Cul5 neddylation was increased following activation in CD4[+] T cells[29]. To monitor the kinetics of Cul5 expression and neddylation in CD4[+] T cells, we assessed Cul5 and neddylated Cul5 levels by immunoblot (IB) in CD4[+] T cells. In resting CD4[+] T cells, we observed two bands representing Cul5 (Fig. 1a). We predicted that the faster-migrating band represented Cul5, while the slower migrating band represented neddylated Cul5. Supporting this, addition of MLN4924, a compound that inhibits neddylation[30], referred to here as NAEi, resulted in an almost complete loss of the upper band and an increase in the lower band. After restimulation of resting cells with anti-CD3 and anti-CD28, the relative proportion of neddylated Cul5 increased, while unneddylated Cul5 decreased compared to resting cells (Fig. 1a).

To evaluate the kinetics of Cul5 expression and neddylation in CD4[+] T cells, we stimulated naïve CD4[+] T cells with anti-CD3 and anti-CD28 and assessed Cul5 at different time points. Using two complementary methods, we found that the overall abundance of Cul5 protein was significantly increased after stimulation[31] (Fig. 1b; Supplementary Fig. 1a). Notably, the neddylated fraction of Cul5 increased to an even greater extent than the unneddylated fraction (Fig. 1b). Interestingly, when the cells were removed from stimulating antibodies and "rested" in IL-2-containing media, the level of Cul5 remained high (Fig. 1b), suggesting that once stimulated, the total abundance of Cul5 was sustained. To test whether Cul5 levels and activation differed in particular subsets of Th cells, we assessed Cul5 neddylation under conditions that promote distinct CD4[+] T cell fates. Cells were cultured under various polarizing conditions (Th0, Th1, Th2, Th17, Treg) and Cul5 levels were determined by IB. In all conditions, Cul5 levels and neddylation were similar (Supplementary Fig. 1b). Taken together, these results indicated that Cul5 is inducibly expressed and activated in CD4[+] T cells regardless of cytokine exposure.

To investigate the biological function of Cul5 in CD4[+] T cells, we generated mice with LoxP sites before exon 9 and after 11 in the Cul5 gene (Fig. 1c). This strategy was employed so that recombination of the LoxP sites would cause a deletion of the exons as well as a frameshift mutation, resulting in a total loss of functional protein. We crossed the resulting Cul5 floxed (Cul5[fl/fl]) mice to animals that expressed CD4-Cre, to generate Cul5[fl/fl]CD4-Cre progeny that lacked Cul5 in all T cells (Fig. 1c). We then assessed Cul5 expression in CD4[+] T cells of the resulting Cul5[fl/fl]CD4-Cre mice by IB (Fig.1d)

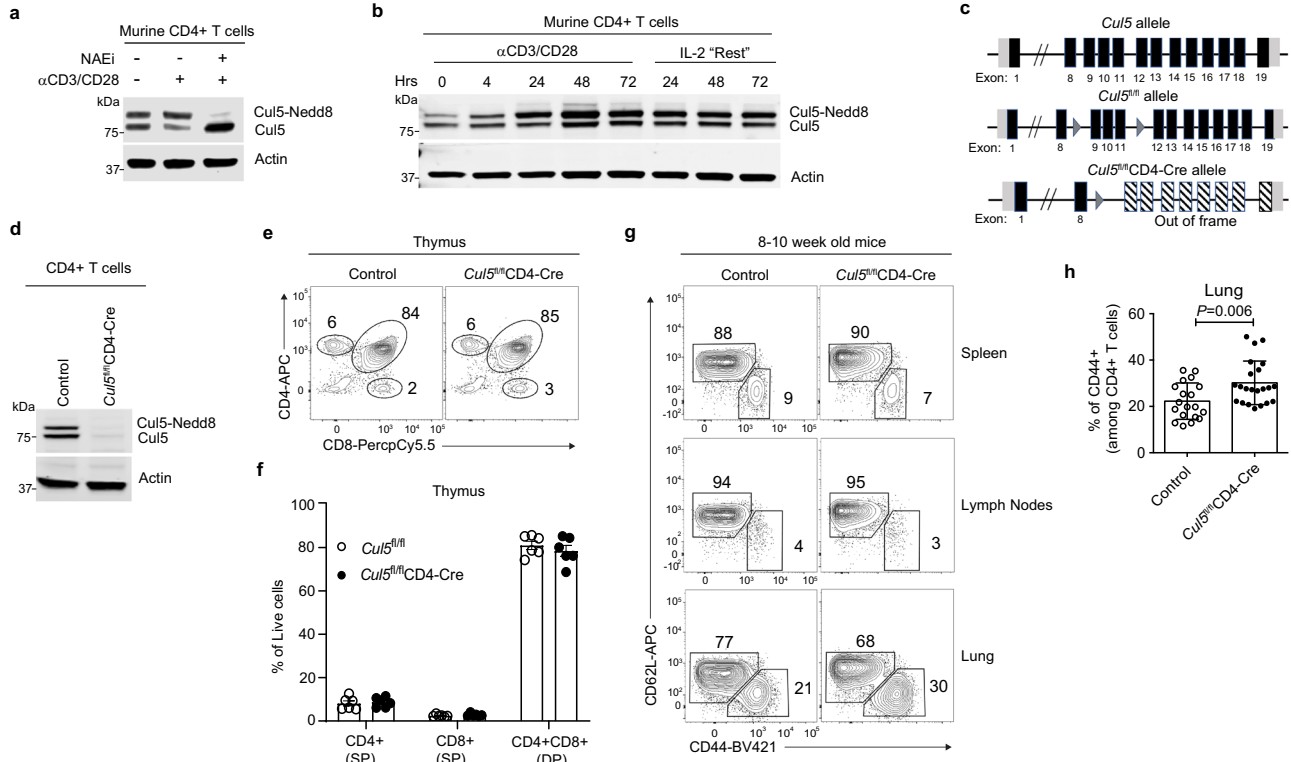

**Fig. 1 Cul5 expression and activation increased following CD4+ T cell stimulation.** Immunoblot analysis showing neddylated and unneddylated forms of Cul5 in CD4+ T cells. **a** CD4+ T cells were stimulated with anti-CD3 and anti-CD28 for 48 h after which cells were washed and kept in IL-2 containing media referred as "Resting media". Resting murine CD4+ T cells were either "stimulated" with anti-CD3 and anti-CD28 for 4hrs or kept in IL-2 containing "resting" media. Nedd8 inhibitor (MLN4924) (NAEi, 1 μM) was added during the final 1 h of restimulation. $n = 3$ biologic replicates were examined in three independent experiments. **b** Naive CD4+ T cells were stimulated with anti-CD3 and anti-CD28 for the indicated time, after 72 h cells were transferred to fresh IL-2 containing media $n = 3$ biologic replicates were examined in three independent experiments. **c** Schematic of *Cul5* gene with loxP insertions. **d** Immunoblot showing expression of Cul5 in CD4+ T cells isolated from spleens of control and *Cul5*fl/fl*CD4-Cre* mice. $n = 3$ biologic replicates were examined in three independent experiments. **e** Representative flow plots showing the frequencies of CD4+ and CD8+ T cells in each thymus. **f** Bar graphs show the compiled data for the frequencies of CD4+ and CD8+ T cells in thymi. $n = 6$ biologic replicates for each genotype were examined in two independent experiments. **g** Representative flow plots showing the frequencies of CD62L+ (Naïve) and CD44+CD4+ T cells from spleens, lymph nodes, and lungs of *Cul5*fl/fl*CD4-Cre* and control mice. **h** Compiled data showing the frequencies of CD44+CD4+ cells from lungs of *Cul5*fl/fl*CD4-Cre* and control mice. $n = 20$ WT and $n = 22$ *Cul5*fl/fl*CD4-Cre* animals were examined in seven independent experiments. Data in panels **f**, **h** are presented as Mean ± SEM. $p$-values were calculated using an unpaired two-tailed $t$-test. Source data are provided in the Source Data file.

and confirmed that Cul5 was deleted. *Cul5*fl/fl*CD4-Cre* mice were born at a normal mendelian frequencies and did not show any overt phenotypic defect. *Cul5*fl/fl*CD4-Cre* mice showed similar frequencies of CD4 and CD8 single positive (SP) cells and double positive (DP) T cells in their thymi (Fig. 1e, f), suggesting that T-cell development was grossly normal in these mice. To examine whether the loss of Cul5 impacted peripheral CD4+ T cell frequencies and/or numbers, we assessed cells in the secondary lymphoid organs and lungs of 8–10-week-old mice using flow cytometry. We found that *Cul5*fl/fl *CD4-Cre* mice had similar frequencies of CD44+CD4+ T cells in their spleens and lymph nodes (Fig. 1g and Supplementary Fig. 1c, d). However, compared to littermate controls, *Cul5*fl/fl*CD4-Cre* mice had higher frequencies and numbers of CD44+ CD4+ T cells in their lungs (Fig. 1g, h, and Supplementary Fig. 1e). Interestingly, we also observed an increase in CD44+CD8+ T cells in the lungs of *Cul5*fl/fl *CD4-Cre* mice compared to controls (Supplementary Fig. 1f).

**Cul5fl/flCD4-Cre mice develop an unprovoked Th2-mediated lung inflammation.** To examine whether the increased CD44+CD4+ T cells in the lung were associated with immunopathology, we first performed histological analysis in control and *Cul5*fl/fl*CD4-Cre* animals. In 8–10-week old mice we did not

observe significant differences in immune infiltration by H&E staining (Fig. 2a, c). Additionally, goblet cell numbers were similar based on PAS staining (Fig. 2a, d). In contrast, 30–36-week-old *Cul5*fl/fl*CD4-Cre* mice showed immune cell infiltration by H&E staining (Fig. 2b, c). In addition, PAS staining showed increased goblet cell hyperplasia compared to age-matched littermate controls (Fig. 2b, d).

To find out whether the infiltrating CD4+ T cells were producing cytokines that promote goblet cell hyperplasia, we examined cytokine production in lung CD4+ T cells. We observed that 8–10-week-old *Cul5*fl/fl*CD4-Cre* mice showed a modest, but reproducible increase in their frequencies of IL-5+ and IL-4+ CD4 cells compared to littermate controls (Fig. 2e, g, h; and Supplementary Fig. 2a, c, d). Their frequencies of IL-13+ cells were elevated but highly variable (Fig. 2e, i, j). By comparison, 30–36-week-old *Cul5*fl/fl*CD4-Cre* mice showed a substantial increase in Th2 cytokine-producing T cells compared to age matched controls (Fig. 2e–j, and Supplementary Fig. 2a–d).

Th2 cytokines can impact the recruitment and/or survival of eosinophils and alveolar macrophages in the lung[32,33]. We found that 30–36-week-old *Cul5*fl/fl*CD4-Cre* mice showed elevated numbers of eosinophils (Fig. 2k, l). However, we did not observe differences in alveolar macrophages in control and *Cul5*fl/fl*CD4-Cre*

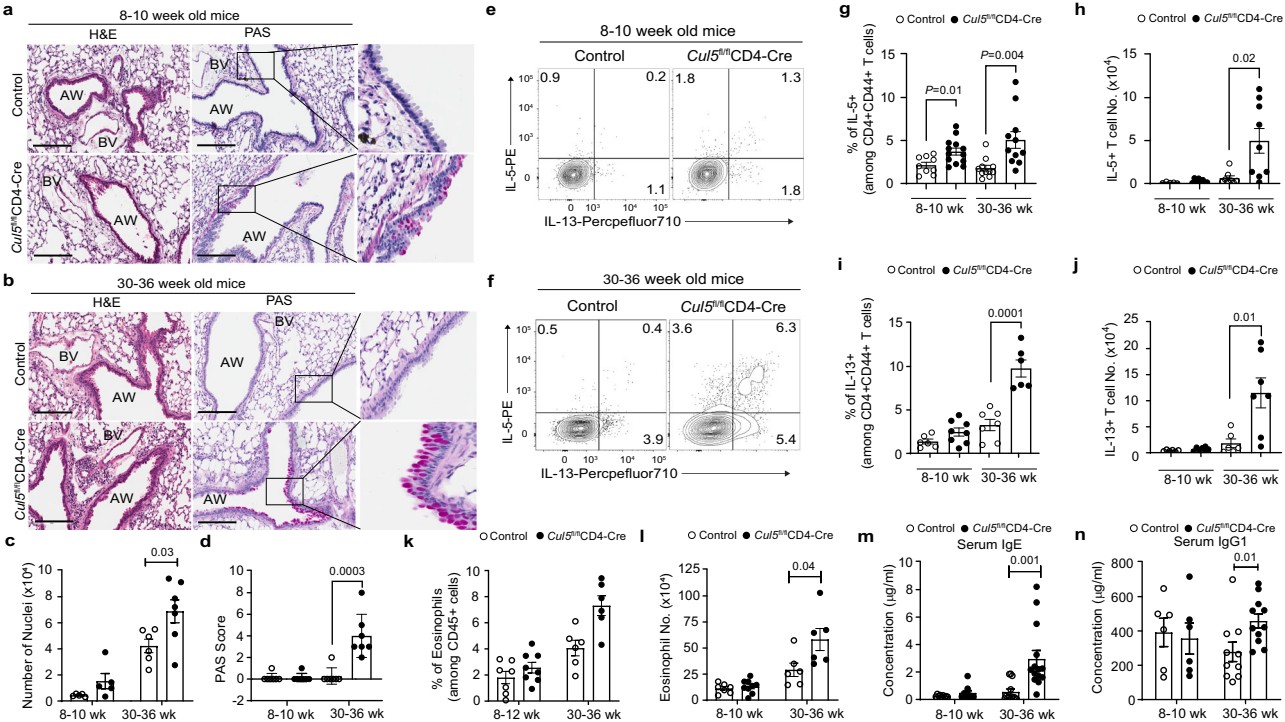

**Fig. 2 *Cul5*<sup>fl/fl</sup>*CD4-Cre* mice develop an unprovoked Th2-mediated lung inflammation.** 8-10 or 30-36 week old control and *Cul5*fl/fl*CD4-Cre* mice were analyzed. Hematoxylin and Eosin (H&E) and Periodic Acid Schiff (PAS) stain of lung sections. Representative image of stains of lung sections from 8-10-week-old (**a**) and 30-36 week old (**b**) mice. Bar graph showing the number of nuclei (**c**) enumerated by Aperio ImageScope software in 100 mm² area of lungs. PAS score (**d**) of the lung sections. For **a** and **c**, n = 5 for 8-10 week old control and *Cul5*fl/fl*CD4-Cre* mice; for 30-36 week old n = 6 control; 7 *Cul5*fl/fl*CD4-Cre* mice. For **b** and **d** n = 7 control and *Cul5*fl/fl*CD4-Cre* mice examined over two independent experiments; Scale bars, 200 μM, zoomed image of PAS is shown in right. **e**, **f** Representative flow plots showing the percentages of IL-5⁺ and IL-13⁺ cells among CD44⁺CD4⁺ T cells from lung. **g** Compiled data showing the frequencies of IL5⁺ cells in 8-10 week old mice n = 9 control; n = 13 *Cul5*fl/fl*CD4-Cre* examined over four independent experiments, and 30-36 week old mice n = 10 control; n = 11 *Cul5*fl/fl*CD4-Cre* examined in 3 independent experiments. **h** Numbers of IL-5⁺ cells in 8-10-week old mice n = 5 control; n = 8 *Cul5*fl/fl*CD4-Cre, examined in three independent experiments, and 30-36-week-old mice, n = 8 control; n = 9 *Cul5*fl/fl*CD4-Cre, examined in two independent experiments. **i** Frequencies and **j** number of IL-13⁺ cells in lungs. For 8-10 week old mice n = 6 control, n = 8 *Cul5*fl/fl*CD4-Cre, examined in 3 independent experiments; for 30-36 week old mice n = 7 control, n = 6 *Cul5*fl/fl*CD4-Cre, examined in two independent experiments. **k** Frequencies of eosinophils in the lungs. For 8-10- week old mice, n = 7 control, 8 *Cul5*fl/fl*CD4-Cre examined in three independent experiments, and for 30-36-week-old, n = 6 mice examined in two independent experiments. **k** Number of eosinophils in the lungs of 8-10- week old mice, n = 7 control, n = 9 *Cul5*fl/fl*CD4-Cre, examined in three independent experiments, and for 30-36-week-old, n = 6 control and *Cul5*fl/fl*CD4-Cre* mice examined in two independent experiments. **m** Serum IgE levels. For 8-10 week old mice, n = 9 control and *Cul5*fl/fl*CD4-Cre* mice examined in three independent experiments, and for 30-36-week-old mice, n = 13 control, n = 14 *Cul5*fl/fl*CD4-Cre, examined over 4 independent experiments. Serum IgG1 levels. For 8-10 week old, n = 12 control and *Cul5*fl/fl*CD4-Cre* mice examined in three independent experiments, and for 30-36-week-old mice, n = 10 control, n = 11 *Cul5*fl/fl*CD4-Cre, examined over three independent experiments. Data is presented as Mean ± SEM in panels **c**, **d**, **g–n**. *p*-value is calculated by unpaired two-tailed *t*-test. AW Airways; BV-blood vessel. Source data are provided as a Source Data file.

animals (Supplementary Fig. 2e). The Th2 cytokine IL-4 can induce class switch recombination (CSR) in B-cells to IgG1 and IgE. While 8–10 weeks old mice did not show differences in serum IgG1 and IgE levels (Fig. 2m, n), 30–36-week-old mice showed significantly increased levels of serum IgG1 and IgE (Fig. 2m, n).

Interestingly, while *Cul5*fl/fl*CD4-Cre* mice showed increased Th2 cells, no significant differences were observed in Th1 cells (IFNγ⁺) (Supplementary Fig. 2f–h) or IL-17A⁺ Th17 cells (Supplementary Fig. 2i–k). In keeping with this, we found no significant differences in the numbers of neutrophils (Supplementary Fig. 2l). Furthermore, we did not observe an increase in mast cells in *Cul5*fl/fl*CD4-Cre* compared to control mice (Supplementary Fig. 2m). Together, these results showed that deletion of Cul5 in CD4⁺ T cells predisposed mice to develop Th2-associated lung inflammation which worsened as they age.

***Cul5*fl/fl*CD4-Cre* mice develop lung remodeling and airway hyperresponsiveness after allergen challenge**. Given that *Cul5*fl/fl *CD4-Cre* mice developed Th2-mediated lung inflammation, we

posited that they might be more susceptible to allergic inflammation. To test this, we challenged 8–10-week-old mice with house dust mite (HDM) extract, a common allergen that drives atopic asthma[34]. *Cul5*fl/fl*CD4-Cre* and littermate control mice were exposed to intranasal PBS or HDM over the course of two weeks (Supplementary Fig. 3a). Histological analysis revealed that HDM treated *Cul5*fl/fl*CD4-Cre* mice developed increased perivascular immune cell infiltration in their lungs compared to control animals (Fig. 3a, b). PAS staining revealed goblet cell hyperplasia in HDM treated *Cul5*fl/fl*CD4-Cre* mice compared to controls (Fig. 3a, c). HDM treated *Cul5*fl/fl*CD4-Cre* mice showed a significant increase in their Bronchoalveolar Lavage (BAL) cells compared to controls (Supplementary Fig. 3b). Further analysis of the BAL cells revealed increased numbers of eosinophils in *Cul5*fl/fl*CD4-Cre* (Fig. 3d) and T cells (Fig. 3e). The majority of the T-cells present in BAL were CD4⁺ T cells, and there was no significant difference in CD8⁺ T-cells (Fig. 3f and Supplementary Fig. 3c). In contrast, neutrophils were not significantly different in the BAL (Supplementary Fig. 3d). It is worth noting that PBS treated

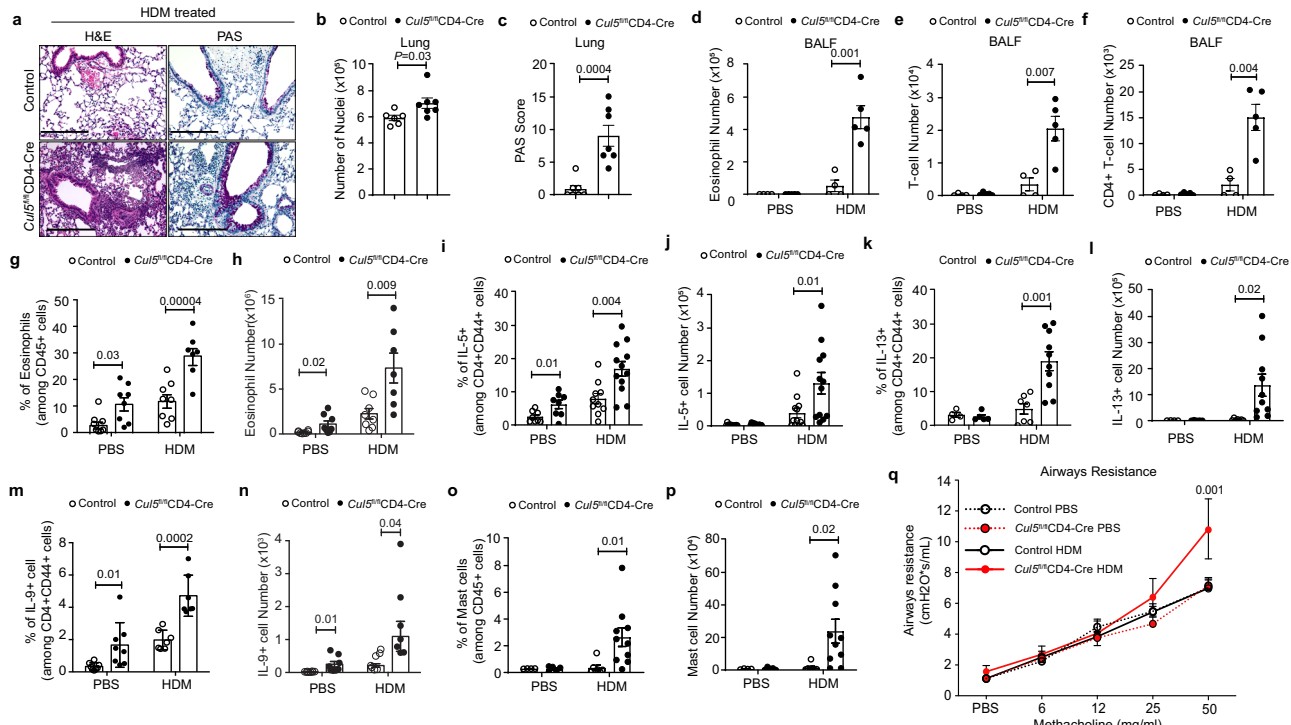

**Fig. 3 Cul5<sup>fl/fl</sup>CD4-Cre mice develop Th9-associated lung remodeling and airways hyperresponsiveness after allergen challenge.** 8–10 week old control and Cul5<sup>fl/fl</sup>CD4-Cre mice were treated with HDM or PBS. Hematoxylin and Eosin (H&E) and Periodic Acid-Schiff (PAS) stain of lungs. Representative image (**a**) of H&E and PAS stain (combined data shown in panels **b** and **c**). Bar graph (**b**) showing the number of nuclei enumerated by Aperio ImageScope software in 100 mm² area. $n = 6$ control, $n = 7$ Cul5<sup>fl/fl</sup>CD4-Cre, examined in two independent experiments. Bar graph showing PAS score (**c**) in lungs. $n = 7$ control and Cul5<sup>fl/fl</sup>CD4-Cre mice, examined in two independent experiments. Total number of eosinophils (**d**), T cells (**e**), and CD4<sup>+</sup> T cells (**f**) in BALF. $n = 4$ control, $n = 5$ Cul5<sup>fl/fl</sup>CD4-Cre examined in 1 experiment. Frequencies (**g**) and numbers (**h**) of eosinophils in the lungs. $n = 7$ for Cul5<sup>fl/fl</sup>CD4-Cre HDM animals; $n = 8$ for all other groups examined in two independent experiments. **i** Percentages of IL-5<sup>+</sup> cells in lungs. For PBS group, $n = 8$ control and Cul5<sup>fl/fl</sup>CD4-Cre mice, examined over two independent experiments. For HDM group $n = 10$ control, $n = 12$ Cul5<sup>fl/fl</sup>CD4-Cre mice, examined in three independent experiments. **j** IL-5<sup>+</sup> cell numbers in lungs. $n = 12$ control and Cul5<sup>fl/fl</sup>CD4-Cre mice examined in three independent experiments. Frequencies (**k**) and numbers (**l**) of IL-13<sup>+</sup> cells. For PBS group, $n = 4$ control, $n = 5$ Cul5<sup>fl/fl</sup>CD4-Cre, examined in one experiment. For HDM group, $n = 7$ control, $n = 11$ Cul5<sup>fl/fl</sup>CD4-Cre, examined in two independent experiments. Frequencies (**m**) and numbers (**n**) of IL-9<sup>+</sup> cells. For PBS group $n = 8$ control and Cul5<sup>fl/fl</sup>CD4-Cre mice. For HDM group $n = 7$ control and Cul5<sup>fl/fl</sup>CD4-Cre mice, examined in two independent experiments. Frequencies (**o**) and numbers (**p**) of mast cells (FcεRIα<sup>hi</sup>c-Kit<sup>hi</sup>). For PBS group $n = 4$ control, $n = 5$ Cul5<sup>fl/fl</sup>CD4-Cre examined in one experiment. For HDM group $n = 7$ control, $n = 11$ Cul5<sup>fl/fl</sup>CD4-Cre, examined in two independent experiments. **q** Airways resistance measured by forced inspiration using Buxco® FinePointe system in mice administered with increasing doses of methacholine. For PBS group $n = 8$ control and Cul5<sup>fl/fl</sup>CD4-Cre mice; for HDM group $n = 7$ control and Cul5<sup>fl/fl</sup>CD4-Cre mice, examined in two independent experiments. Data is presented as Mean ± SEM in panels **b**–**q**. p-value is calculated by unpaired two-tailed t-test. Source data are provided as a Source Data file.

Cul5<sup>fl/fl</sup>CD4-Cre and control mice showed similar number of cells in their BAL (Supplementary Fig. 3b), supporting that Cul5<sup>fl/fl</sup>CD4-Cre mice showed little to no inflammation prior to HDM treatment. Analysis of the cytokine profiles of BAL fluid by ELISA show increased IL-4, IL-6 and monocyte chemoattractant protein 1 (MCP-1) in HDM treated Cul5<sup>fl/fl</sup>CD4-Cre mice compared to controls (Supplementary Fig. 3e).

We then analyzed cells in the lungs using flow cytometry. Cul5<sup>fl/fl</sup>CD4-Cre mice had increased frequencies and numbers of eosinophils in their lungs after HDM exposure compared to control animals (Fig. 3g, h). This suggested that there was an increased Th2-mediated inflammation in the lungs of Cul5<sup>fl/fl</sup>CD4-Cre mice following HDM exposure. Supporting this, when we assessed cytokines produced by antigen-experienced CD4<sup>+</sup>CD44<sup>+</sup> cells, we found that Cul5<sup>fl/fl</sup>CD4-Cre mice showed significant increases in their frequencies and numbers of IL-5<sup>+</sup> cells compared to controls (Fig. 3i, j). Cul5<sup>fl/fl</sup>CD4-Cre mice also showed increased IL13<sup>+</sup> cells compared to control mice (Fig. 3k, l). We did not observe significant changes in the frequencies of IL-4<sup>+</sup> cells (Supplementary Fig. 3f). We did not observe significant changes in the numbers of

IL-4<sup>+</sup> cells in PBS treated group, this could be due to presence of saline in lungs which has been shown to change the recruitment and function of immune cells[48,49].

Previous studies have suggested that Th9 cells increase following allergen exposure in mice and in asthmatic patients[20,35]. These cells are thought to contribute to the pathophysiology of asthma. We found that IL-9<sup>+</sup> T cells were significantly increased in Cul5<sup>fl/fl</sup>CD4-Cre mice compared to controls (Fig. 3m, n). IL-9 promotes the survival and proliferation of mast cells[1,5]. Consistent with this, we found elevated frequencies and numbers of mast cells in Cul5<sup>fl/fl</sup>CD4-Cre mice compared to control mice (Fig. 3o, p, and Supplementary Fig. 3g). Given that Th9 cells are generated in the presence of IL-4 and TGF-β, we assessed TGF-β levels in the lungs of HDM-treated mice. We found similar levels of TGF-β in the BALF of Cul5<sup>fl/fl</sup>CD4-Cre and control animals (Supplementary Fig. 3h).

To further characterize the immune response upon HDM exposure we measured Th1 (IFNγ<sup>+</sup>) and Th17 (IL17A<sup>+</sup>) cells in lung by flow cytometry. We observed a trend towards higher numbers of Th1 and Th17 cells (Supplementary Fig. 3i, j). We did

not observe an elevation of HDM-specific IgE levels in HDM exposed Cul5[fl/fl]CD4-Cre mice (Supplementary Fig. 3k). This is in keeping with a previous study that found that acute HDM exposure was not sufficient to induce HDM-specific IgE[34].

Finally, to test whether increased inflammation in the lungs of Cul5[fl/fl]CD4-Cre animals impacted lung function, we measured airways resistance by administering methacholine and analyzing the forced inspiratory capacity via Buxco FinePointe. We observed that control animals showed similar airways resistance after exposure to HDM or PBS (Fig. 3p). This is consistent with previous studies showing that two-week HDM exposure is not sufficient to induce airways resistance in C57BL/6 mice[34]. In contrast, Cul5[fl/fl]CD4-Cre mice showed increased airways resistance (Fig. 3p), suggesting that Cul5[fl/fl]CD4-Cre mice were more susceptible to HDM-induced asthma. Taken together, these results indicated that Cul5 limits the frequencies and numbers of Th2 and Th9 cells, prevents lung remodeling and other pathologic features associated with asthma.

**CIS acts as a substrate receptor for CRL5.** Having determined that Cul5 deficiency in CD4+ T cells predisposes mice to Th9-mediated lung inflammation, we next wanted to assess how Cul5 functions in CD4+ T cells. Cul5 is a scaffold protein that assembles the CRL5 complex. Substrate specificity of the CRL5 complex depends on recruitment of a substrate receptor[24,26]. SOCS family proteins can act as substrate receptors for the CRL5 complex[10]. To identify the substrate receptor for Cul5 in CD4+ T cells, we immunoprecipitated (IP) Cul5 in primary mouse CD4+ T cells and identified associated proteins using tandem mass spectrometry (IP-MS/MS). Given that Cul5-deficiency resulted in increased numbers of both Th2 and Th9 cells, and that these cells require IL-4 for their differentiation[36–38], we assessed Cul5 associated substrate receptors in the presence and absence of exogenous IL-4. We identified several known components of the CRL5 complex, such as EloB/C, DCN1, and CAND1. Importantly, we also identified two SOCS box family proteins, CIS and Suppressor of Cytokine Signaling 2 (SOCS2) (Fig. 4a). Although both CIS and SOCS2 have been shown to limit Th2 differentiation[17,20], only CIS has been shown to restrict the differentiation of both Th2 and Th9 cells[20]. Notably, we found that the amount of CIS co-precipitated with Cul5 was higher than for SOCS2 (Fig. 4a). This result, combined with the finding that Cul5[fl/fl]CD4-Cre mice have phenotypic similarities to CIS-/- animals[20], we posited that CIS might act as a substrate receptor for Cul5 in CD4+ T cells.

To further assess whether CIS associates with Cul5, and participates within a CRL5 complex, we performed Cul5 IPs followed by CIS IB. We identified CIS in the Cul5 IP but not in the IgG control (Fig. 4b). To further define the signals that promote an association between Cul5 and CIS, we employed the murine Th2 derived D10 cell line. Immunoblot analysis showed that Cul5 and CIS were both expressed in D10 cells, and that CIS expression increased upon anti-CD3 and anti-CD28 stimulation (Fig. 4c, d). To test whether CIS was part of a CRL5 complex in D10 cells, we immunoprecipitated CIS and identified its interacting partners using a similar IP-MS/MS approach as described above. We identified Cul5, EloB/C, CAND1, and multiple components of the Cop9 signalosome (CSN) (Fig. 4e), all proteins shown to be part of a CRL5 complex[24]. Given that our IP-MS/MS results in Fig. 4a supported that Cul5 and CIS could associate even in the absence of Cul5 neddylation, we next sought to test this in D10 cells using Cul5 IP and CIS IB (Fig. 4f). Again, Cul5 and CIS association was detected in cells that were treated with the NAEi. Taken together these results supported that CIS acts as a substrate receptor in a CRL5 complex.

We next evaluated signals that promote CIS expression, as this may be the limiting factor in CRL5 formation. Given that, Cul5 deficiency resulted in increased Th2 and Th9 cells, we tested whether IL-4R signaling regulates CIS protein levels. While CIS protein levels were below the limit of detection in unstimulated cells, similar levels were detected in anti-CD3 and anti-CD28 stimulated regardless of whether or not they were exposed to exogenous IL-4 or anti-IL-4, to deplete IL-4 in the culture media (Fig. 4g, h). The efficiency of IL-4 blockade was assessed by pSTAT6 levels. Anti-IL-4 treated cells showed almost complete loss of pSTAT6. We also performed similar experiments in primary murine CD4+ T cells. Here again CIS protein levels were increased in stimulated cells regardless of IL-4 or anti-IL-4 treatment (Fig. 4i, j). These results suggested IL-4 is not the only signal capable of driving CIS expression. This was further supported by the observation that CIS protein levels were similar under differentiation conditions that do or do not include exogenous IL-4, namely Th9 and Th0 conditions (Fig. 4k).

**Cul5-deficient CD4+ T cells show an increased Th9 gene signature.** Previous studies have shown that CD4+ T cells lacking CIS have an increased ability to become Th2 and Th9 cells[20]. We posited that if CIS functions as a substrate receptor for Cul5, T cells lacking Cul5 should also have an increased propensity to become Th2 and Th9 cells. Thus, we isolated naïve CD4+ T cells from WT and Cul5[fl/fl]CD4-Cre mice and cultured them under Th1, Th2, Th9, and Th17 polarizing conditions. We found that Cul5-deficient CD4+ T cells were more likely to become Th2 and Th9 cells (Fig. 5a, b), conditions in which IL-4 was added to the cultures. On the other hand, Cul5-deficient CD4+ T cells differentiated into Th1 cells at similar frequencies as controls (Supplementary Fig. 5a). Interestingly, Cul5 deficient CD4+ T cells showed slightly decreased frequencies of Th17 cells (Supplementary Fig. 5b).

To gain insight into the mechanism by which Cul5 regulates Th9 differentiation we assessed changes in gene expression by RNAseq in WT and Cul5-deficient CD4+ T cells cultured under Th9 conditions. To use an unbiased approach to determine how Cul5-deficiency regulates Th differentiation we used a publicly available dataset from a study that identified genes associated with different Th subtypes[39]. We selected the top 200 genes for each Th subtype by identifying the most differentially expressed genes compared to naïve CD4+ T cells (Log2 Fold change > 2), referred as 'top hits'. We then compared the relative expression of these genes in WT and Cul5-deficient (Cul5 cKO) cells. We found that Cul5-deficient cells were most significantly enriched for genes associated with Th9 cells compared to other T helper subtypes (Fisher exact test, adj p-value < 0.05) (Fig. 5c). Next, to identify changes in the gene expression we compared differentially expressed genes (DEGs) (shown in Fig. 5c), between WT and Cul5 cKO CD4+ T cells with the 'top hit' genes identified in different Th subtypes described above. This analysis would indicate if Cul5 regulates gene expression in a particular Th subtype. We identified 29 DEGs associated with the Th9 signature of which 25 had higher expression and 4 had lower expression in Cul5-deficient cells compared to WT (Fig. 5d). Fewer differentially expressed genes were identified for Th2 (17 genes), Th1 (18 genes), Th17(19 genes) and Th0 (16 genes) (Fig. 5e, f and Supplementary Fig. 5c, d). While Th1 and Th17 'top hits' also had a comparably high ratio of upregulated DEGs to downregulated DEGs, because of their lower quantity of upregulated DEGs compared to Th9 and Th2 'top hit' genes, there was lower enrichment of Cul5 cKO DEGs in the Th1 and Th17 'top hits' than the Th9 or Th2 'top hits' as seen by the lower p-values (Fig. 5c). Interestingly, most of the genes found in other subtypes

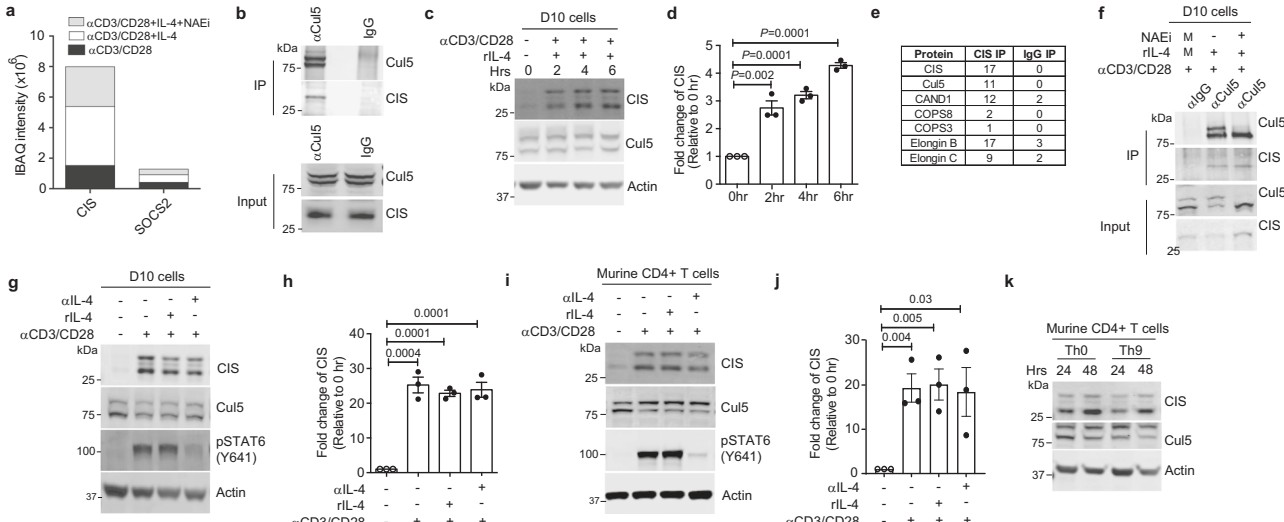

**Fig. 4 CIS (Cytokine inducible SH2 protein) acts as a substrate receptor for CRL5. a** IBAQ intensities for CIS and SOCS2 proteins. CD4+ T cells from 8–10-week-old mice were restimulated for 4 h with either anti-CD3 and anti-CD28, or anti-CD3 and anti-CD28 in the presence of IL-4, or anti-CD3 and anti-CD28 in the presence of IL-4 and treated with Neddylation inhibitor (NAEi) for the final one hour. After the respective treatment, the lysates were immunoprecipitated by anti-Cul5 or IgG antibodies, and samples were analyzed by mass spectrometry. $n = 2$ biologic replicates of cells examined in two independent experiments. **b** Immunoblot showing interaction of Cul5 and CIS in primary CD4+ T-cells. $n = 2$ biologic replicates of cells examined in two independent experiments. Representative Immunoblot (**c**) and fold change in CIS expression (**d**) in D10 cells. $n = 3$ biologic replicates of cells examined in three independent experiments. To calculate fold change, the normalized CIS level at 0 h was adjusted to 1 and the relative change in the expression of CIS protein was calculated accordingly. **e** Table showing components of the CRL5 E3 ubiquitin complex that were identified by CIS IP and IgG IP in D10 cells. Cells were stimulated in the presence of IL-4 and NAEi for 4 h. Total number of peptides identified in IP is listed in the table. $n = 2$ biologic replicates of cells examined in two independent experiments. **f** Immunoblot of Cul5 IP showing its interaction with CIS in D10 cells. $n = 3$ biologic replicates of cells examined in three independent experiments. Immunoblot (**g**) and fold change (**h**) of CIS in D10 cells stimulated in the presence of exogenous IL-4 or anti-IL-4 for 4 h. $n = 3$ biologic replicates of cells examined in three independent experiments. Immunoblot (**i**) and fold change of CIS protein (**j**) in murine CD4+ T-cells isolated from 8–10-week-old mice. $n = 3$ biologic replicates of cells examined in three independent experiments. **k** Immunoblot showing expression of CIS protein in murine CD4+ T cells. $n = 3$ biologic replicates of cells examined in three independent experiments. Data is presented as Mean ± SEM in panels **d**, **h** and **j**. p-value was calculated using unpaired two tailed t-test. Source data are provided as Source Data file.

were not unique (Fig. 5d, f and Supplementary Fig. 5c, d), indicating that these genes are likely upregulated in response to T-cell activation. Furthermore, we compared our dataset with another publicly available dataset where authors identified Th9-associated genes using a microarray[40]. Here again, we observed that Cul5 cKO cells showed an increase in Th9-associated genes compared to control cells (Supplementary Fig. 5e).

The increase in enrichment of Th9 genes in Cul5-deficient CD4+ T cells led us to hypothesize that Cul5 might be regulating a transcription factor(s) involved in Th9 differentiation. To test this, we identified 'top hit' DEGs that were unique to Th9 cells and then sought to identify transcription factors that bound to the unique genes. We identified 6 genes (Il9, Batf3, ppp2r3a, Adarb1, Nek6, Gpm6b) that are specifically enriched in Th9 cells (Fig. 5g). To assess transcription factor binding, we used publicly available ChIP-seq data. Given that Cul5-deficient T cells have increased production of IL-9, we limited ourselves to transcription factors known to regulate IL-9 expression (STAT6, STAT5, BATF, PU.1 and IRF4)[35]. ChIPseq data supported that STAT6 associates with 5 of these Th9 unique genes (Il9, Batf3, ppp2r3a, Adarb1, Nek6), while others associated with three or fewer genes (Fig. 5h). These results suggested that Cul5 limits Th2- and Th9 differentiation by limiting STAT6 transcriptional activity.

**CRL5 ubiquitinates and degrades pJak1.** To evaluate whether Cul5 limits STAT6 transcriptional activity, we cultured CD4+ T cells in vitro, to allow CIS levels to increase, and then stimulated them with IL-4 and measured pSTAT6. Indeed, we observed that Cul5-deficient CD4+ T cells had elevated levels of pSTAT6

compared to WT cells by IB and flow cytometry respectively (Fig. 6a, b, and Supplementary Fig. 6a, b). This is similar to published studies from CD4+ T cells lacking CIS[20]. One possibility for the increased pSTAT6 in Cul5 deficient CD4+ T cells is that Cul5 and CIS interact with pSTAT6 directly and promote its ubiquitination. To test this, we performed a CIS IP in D10 cells treated with IL-4 in combination with NAEi (to prevent substrate ubiquitination and degradation) and identified associated proteins using IP-MS/MS. While we did not identify STAT6, we found both Jak1 and IL-4Rα (Fig. 6c). This was particularly noteworthy because Jak1 and IL-4Rα are upstream of STAT6 phosphorylation[41]. Thus, we posited that Cul5 and CIS might target either Jak1 or IL-4Rα. Given that CIS has an SH2 domain that allows it to recruit phosphorylated substrates to the Cul5 complex, we specifically focused on pJak1 and pIL-4Rα. To test this, we used the NAEi to block CRL5 neddylation in D10 cells. D10 cells were stimulated with anti-CD3 and anti-CD28 in the presence of IL-4 alone, or IL-4 with NAEi. Levels of pJak1 and pIL4Rα were assessed by IB. We found that NAEi treatment resulted in elevated levels of pJak1 and pIL4Rα compared to untreated cultures (Fig. 6d and Supplementary Fig. 6c). These data supported that Cul5 regulates the levels of pJak1 and/or pIL-4Rα.

To directly test whether Cul5 regulates pJak1 levels, we compared pJak1 levels in WT and Cul5-deficient CD4+ T cells. Cul5-deficient CD4+ T cells showed higher pJak1 and pIL-4Rα levels compared to WT cells (Fig. 6e and Supplementary Fig. 6d). This difference was particularly evident at later time points. Next, to analyze pJak1 stability, we treated WT and Cul5-deficient CD4+ T cells with cycloheximide (CHX) to block new protein

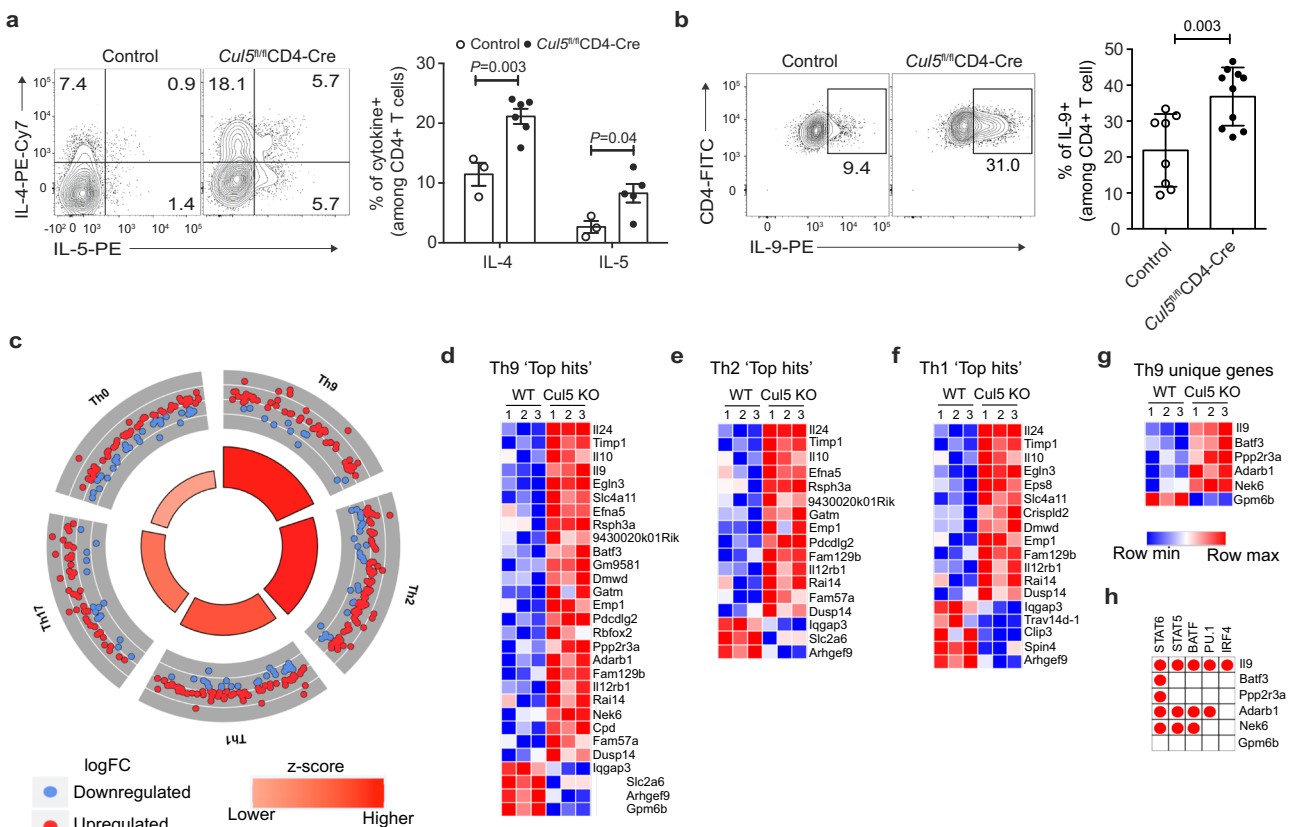

**Fig. 5 Cul5-deficient CD4$^+$ T cells show an increased Th9 gene signature.** Naïve CD4$^+$ T cells from 8–10-week-old control and *Cul5$^{fl/fl}$CD4-Cre* mice were isolated and cultured in Th2 and Th9 polarising conditions. **a** Left panel shows the representative flow plot for IL-4$^+$ and IL-5$^+$ cells under Th2 polarizing condition. The bar graph on right shows the consolidated data. For IL-4$^+$ cells, $n = 3$ control, $n = 6$ *Cul5$^{fl/fl}$CD4-Cre*, for IL-5$^+$ cells, $n = 3$ control, $n = 5$ *Cul5$^{fl/fl}$CD4-Cre*, examined in two independent experiments. **b** Left panel shows the representative flow plot for IL-9$^+$ cells in control and *Cul5$^{fl/fl}$CD4-Cre* CD4$^+$ T cells cultured under Th9 polarizing conditions. The bar graph on the right shows the consolidated data for $n = 8$ control, $n = 10$ *Cul5$^{fl/fl}$CD-Cre*, examined in four independent experiments. **c–h** RNASeq analysis was used to measure the transcript levels in control and Cul5 deleted CD4$^+$ T cells cultured for 2 days in Th9 polarizing condition. $n = 3$ control and *Cul5$^{fl/fl}$CD4-Cre* CD4$^+$ T cells examined in one experiment. **c** Ontological analysis of the differentially regulated genes, calculated by GOplot algorithm is shown. Circle plots depict the enrichment of genes associated with particular Th subtype. Greater the size of the inner trapezoids represents greater -log10 value (adj. *p*-value). Heatmap of Th subtypes. To generate heat maps, we first selected genes that were enriched in different Th subtype compared to naïve CD4 T cells from publicly available dataset[39]. We selected 200 most differentially regulated genes for each subtype and classified these as top hit genes. We then compared these top hits genes to the differentially regulated genes from our RNAseq dataset and to generate heatmaps. Heatmap of top hit genes identified in Th9 (**d**), Th2 (**e**), and Th1 (**f**) subtypes that were also differentially regulated in WT and Cul5 deficient CD4$^+$ T cells. **g** Heatmap of top hit genes which were identified exclusively in Th9 subtype. **h** The red circle in the table indicates the binding sites for the respective transcription factors present in the particular gene. Data is presented as Mean ±SEM in panels **a**, **b**. *p*-value was calculated using unpaired two tailed *t*-test. Source data are provided as a Source Data file.

synthesis and allow analysis of protein degradation. We found that pJak1 was significantly more stable in Cul5-deficient cells than in WT cells (Fig. 6f). We reasoned that if pJak1 is a substrate of Cul5, we should see pJAK1 interacting with Cul5 and its ubiquitylation should be reduced when Cul5 activity was inhibited. To test whether pJak1 interacts with Cul5, we immunoprecipitated Cul5 and assessed pJak1 association via IB. We found that pJak1 co-precipitated with Cul5 (Fig. 6g) in D10 cells and that this association did not depend on Cul5 neddylation. We next assessed the ubiquitination of pJak1 in the presence or absence of NAEi, to block Cul5 activity. We enriched for ubiquitinated proteins in cell lysates using TUBEs (tandem-ubiquitin binding entities) and then added deubiquiti-nating enzyme to half of the eluate to cleave the ubiquitin chains and release the substrate[29]. Following TUBE enrichment, ubiquitylation of pJAK1 was reduced in cells in which Cul5 was inactivated by NAEi, (Fig. 6h, lanes 3 and 4). DUB treatment resulted in a complete loss of higher molecular species, supporting that these species represented ubiquitylated pJak1

(Fig. 6h, lanes 3 and 5). Furthermore, in both cases the intensity of band corresponding to pJak1 was lower in NAEi treated cells (Fig. 6h, Lane 4 and 6), supporting that pJak1 was less ubiquitylated when Cul5 activity is inhibited. These results supported that CIS and Cul5 regulate the ubiquitination and degradation of pJak1.

**Cul5 regulates fate choice in CD4$^+$ T cells.** During differentiation, the ability of T cells to upregulate lineage-determining transcription factors is based on the types of cytokine signals they receive[1,5]. For example, CD4$^+$ T cells cultured in TGF-β and IL-2 upregulate FoxP3 and become tolerogenic Treg cells. In contrast, addition of IL-4 to these cultures inhibits FoxP3 and promotes GATA3, IRF4 and BATF to allow the generation of proinflammatory Th9 cells[42]. Given that Cul5 deficient CD4$^+$ T cells have increased IL-4R signaling, we hypothesized that Cul5 might regulate fate choice during the differentiation of Treg and Th9 cells. To test this, we cultured naïve WT and Cul5-deficient CD4$^+$

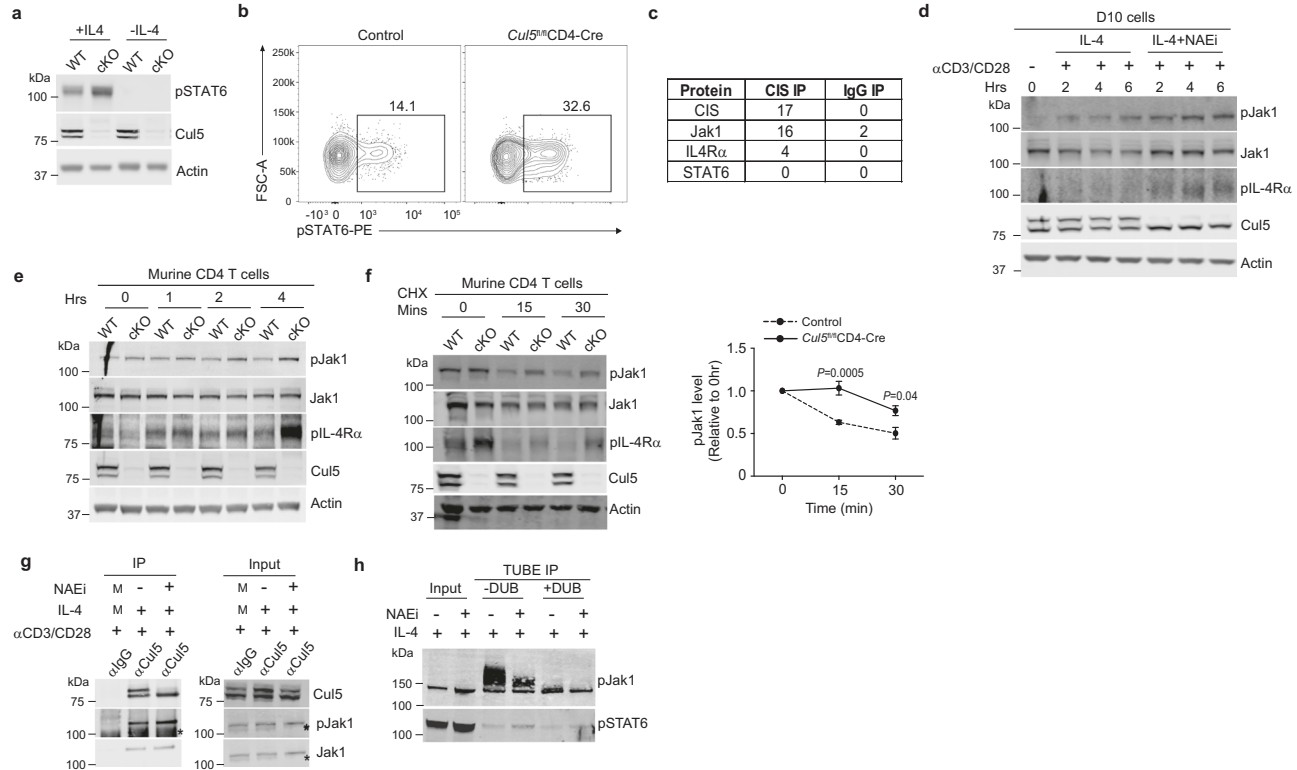

**Fig. 6 CRL5 ubiquitinates and degrades pJak1.** CD4+ T cells isolated from 8–10 week old mice were stimulated for 24hrs washed and treated with 0.25 ng/ml IL-4 for 5 min in plain RPMI media. **a** Immunoblot showing pSTAT6 levels in control and *Cul5*fl/fl*CD4-Cre* CD4+ T cells. n = 3 biologic replicates of cells examined in two independent experiments. **b** Flow plots showing pSTAT6 level in control and *Cul5*fl/fl*CD4-Cre* CD4+ T cells by flow cytometry. n = 4 biologic replicates of cells examined in two independent experiments. **c** Table showing components of the IL-4 signaling pathway co-precipitated with CIS IP or IgG IP in D10 cells. n = 2 biologic replicates of cells examined in two independent experiments. **d** Immunoblot in D10 cells treated with IL-4 or IL-4+NAEi for the indicated time. n = 3 biologic replicates of cells examined in three independent experiments. **e** Immunoblot of control and Cul5 deficient CD4+ T cells isolated from 8–10-week-old mice. Cells were treated with IL-4 for the indicated time. n = 3 biologic replicates of cells examined in three independent experiments. **f** Immunoblot of WT and Cul5 deficient CD4+ T cells stimulated with anti-CD3 and anti-CD28 for 48 h in the presence of anti-IL-4. Cells were then washed, stimulated with IL-4 for 2 h and washed again, and treated with anti-IL-4 along with CHX for the indicated time. Graph on the right shows pJak1 levels upon CHX treatment relative to 0 h. n = 3 biologic replicates of cells examined in three independent experiments. **g** Immunoblot of Cul5 IP in D10 cells. n = 3 biologic replicates of cells examined in three independent experiments M: Similar amounts of lysate from all conditions was pooled for that sample. **h** Representative immunoblot of TUBE immunoprecipitation. D10 cells were stimulated with anti-CD3 and anti-CD28 for 4 h in the presence of IL-4 or IL-4 and NAEi. Cells were lysed and subjected to TUBE IP. The enriched fraction was divided into two parts, one part was treated with DUB and other was left untreated. 3% lysate was used for input (n = 2 biologic replicates of cells examined in two independent experiments). Data is presented as mean ± SEM in panel **f**. p-value was calculated by unpaired two tailed t test. Source data are provided as a Source Data file.

T cells in the presence of TGF-β and IL-2 with increasing concentrations of IL-4. We found that both WT and Cul5-deficient CD4+ T cells showed similar frequencies of FoxP3+ T cells when IL-4 was not added to the cultures, and IL-9+ cells were not observed under these conditions (Fig. 7a, b). The addition of IL-4 to the cultures resulted in fewer FoxP3+ T cells in both genotypes. However, Cul5-deficient cell showed a pronounced decrease in FoxP3+ T cells at significantly lower concentrations of IL-4 than their WT counterparts (Fig. 7a, b). As IL-4 was added to the cultures, IL-9+ T cells were observed in the same cultures as the FoxP3+ cells, and their frequencies increased with higher concentrations of IL-4. The frequencies of IL-9+ cells in the cultures were significantly higher among Cul5-deficient cells (Fig. 7a, b). Under these culture conditions, we did not detect any IL-4+ cells (Supplementary Fig. 7a). These data supported that Cul5 regulates fate decisions at the interface between Treg and Th9 cell differentiation, thus driving the development of tolerogenic FoxP3+ Treg cells rather than proinflammatory Th9 cells. To test this in vivo we treated WT and *Cul5*fl/fl*CD4-Cre* mice with either PBS (Fig. 7c) or HDM (Fig. 7d) for 2 weeks and analyzed cells isolated from the lungs. We found that *Cul5*fl/fl*CD4-Cre* mice

exposed to HDM showed reduced frequencies of FoxP3+ Treg cells in their lungs compared to HDM exposed controls.

To test this further, we generated mixed bone marrow chimeras by reconstituting Rag1-deficient mice with a 1:1 mixture of congenially distinct *Cul5*fl/fl*CD4-Cre* (CD45.2+) and WT (CD45.1+) bone marrow. After 7–8 weeks of reconstitution, we exposed mice to HDM (Supplementary Fig. 7b). This experiment allowed us to examine Cul5-deficient and -sufficient cells in the same cytokine milieu and test the impact of the loss of Cul5 within individual CD4+ T cells. We first evaluated reconstitution by comparing the proportion of B cells from each donor, reasoning that these are Cul5-sufficient in both cases. CD45.1+ and CD45.2+ B cells were similarly reconstituted (Supplementary Fig. 7c). Total CD4+ T cell were also similarly reconstituted among recipients (Supplementary Fig. 7d). Among CD4+ T cells, the ratio of naïve (CD62Lhi) and activated (CD44hi) populations were also similar (Supplementary Fig. 7e, f). We then compared the relative proportion of Foxp3+ Tregs cells that were Cul5-sufficient and Cul5-deficient. We found that Cul5-deficeint CD4+ T cells were less likely to be FoxP3+ Treg cells compared to Cul5-sufficient cells (Fig. 7e).

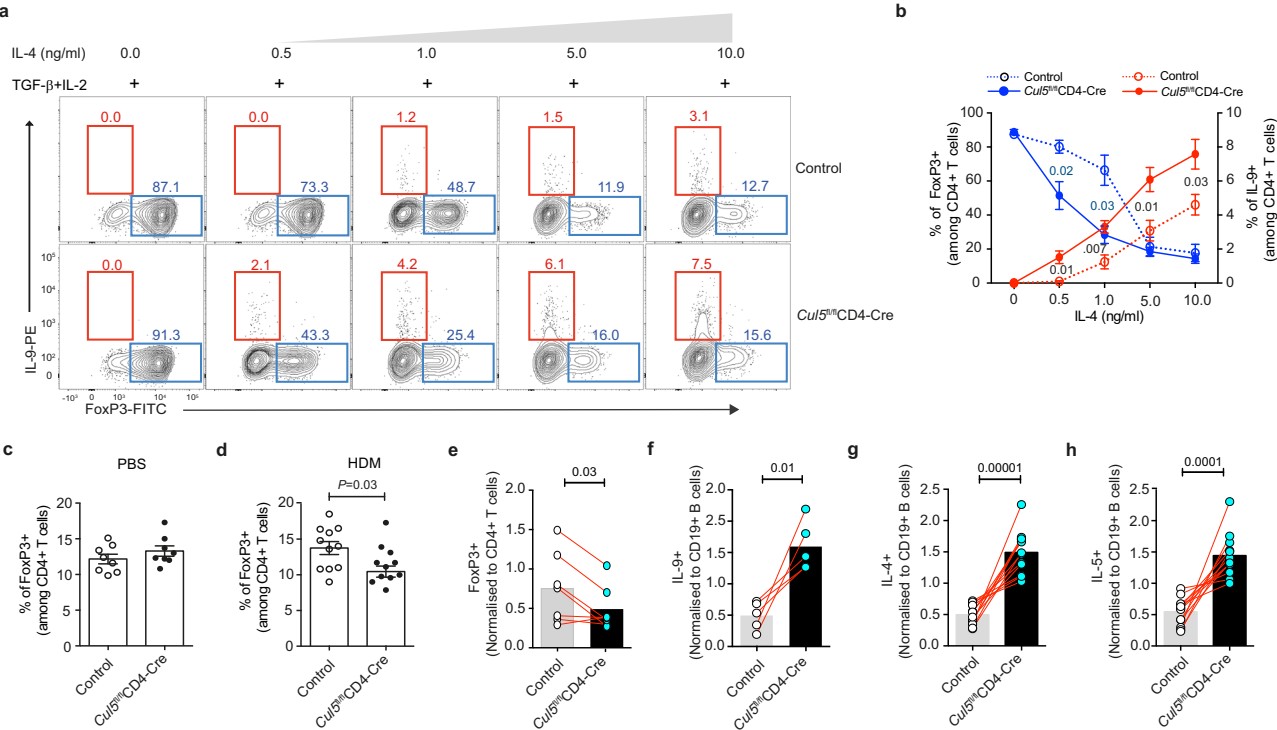

**Fig. 7 Cul5 regulates fate choice in CD4+ T cells. a** Representative flow plots showing the frequencies of FoxP3+ and IL-9+ cells. Naïve CD4+ T cells from control and *Cul5fl/flCD4-Cre* mice were cultured in iTreg polarizing (TGF-β + IL2 + anti-IFNγ) condition with increasing concentrations of IL-4. Cells were stained for FoxP3 and IL-9 on day 5. **b** Compiled data of (**a**) showing the frequencies of FoxP3+ (blue) and IL-9+ (red) cells. For FoxP3+ cells, n = 3 control, n = 4 *Cul5fl/flCD4-Cre*, for IL9+ cells n = 4 control, n = 5 *Cul5fl/flCD4-Cre* biologic replicates of cells examined in three independent experiments. **c** Compiled data showing the frequencies of FoxP3+ cells in 8–10-week-old control and *Cul5fl/flCD4-Cre* mice treated with PBS. n = 8 control and *Cul5fl/flCD4-Cre* mice examined in two independent experiments. (**d**) HDM treated mice n = 11 control and *Cul5fl/flCD4-Cre* mice examined in three independent experiments. Mixed bone marrow chimeras. 6–8-weeks-old Rag1-/- deficient recipients were sublethally irradiated and then injected with a 1:1 mixture of control (CD45.1+) and *Cul5fl/flCD4-Cre* (CD45.2+) donor bone marrow cells and allowed to reconstitute for 8 weeks. These mice were then treated with a 2-week regimen of house dust mite (HDM), and their lung cells were analyzed by flow cytometry. **e** Proportion of FoxP3+ cells that came from each donor. The percent of control (CD45.1+) and *Cul5fl/flCD4-Cre* (CD45.2+) FoxP3+ cells was normalized to the percent of CD4+ T cells from the same donor. n = 7 recipient mice were examined in two independent experiments. Two donors of each genotype were used to reconstitute these recipients. Each recipient received cells from one mouse of each genotype. Proportion of cytokine producing cells. To calculate the contribution from each donor CD45.1 "cytokine+" cells or CD45.2 "cytokine+" cells were normalized to CD19+ cells. **f** Proportion of IL-9+ cells from CD45.1 (WT) and CD45.2 (Cul5 deficient), n = 5 recipients were examined in one experiment. **g-h** Proportion of IL-4+ and IL-5+ cells, n = 12 recipients were examined in two independent experiments. Data is presented as mean±SEM in panels **b–d**. In panels **e-h** data is represented as mean. p-value was calculated by unpaired two tailed t-test used in panels **b-d** and paired two-tailed t-test in panels **e-h**. Source data are provided as a Source Data file.

To assess cytokine production, we examined the CD4+CD44+ T cells that produced IL-4, IL-5, IL-13, IL-9, and IL-17A, and assessed the relative proportions of these cytokine-producing cells that were Cul5-sufficient and Cul5-deficient (Supplementary Fig. 7g). To ensure that the differences in cytokine-producing cells were not due to differences in reconstitution, we normalized the CD45.1 or CD45.2 "cytokine+" cells to CD45.1+ or CD45.2+ B cells within each recipient. We found that Cul5 deficient cells were more likely to be IL-9 producers compared to Cul5 sufficient cells (Fig. 7g). Furthermore, Cul5 deficient cells were more likely to become Th2 cells indicated by IL-5, IL4, and IL-13 staining. (Fig. 7g, h, and Supplementary Fig. 7h). This was not the case for all cytokine-producing CD4+ effector cells, as we did not observe any significant differences among IL-17A+ (Th17) cells (Supplementary Fig. 7i). Together these results indicated that Cul5 acts in CD4+ T cells to regulate fate choice, promoting the differentiation of tolerogenic Treg cells and limiting the differentiation of Th9 cells.

## Discussion

The differentiation of CD4+ T cells into different Th subtypes affords protection against a diverse range of pathogens[1–3]. How a

CD4+ T cells decides its fate in vivo in the presence of multiple cytokines with varied abundance is not understood. Here we show that Cul5 aids fate selection by setting a threshold for IL-4 receptor signaling. In doing so, it facilitates Treg lineage selection, while reducing the potential to become a proinflammatory Th9. This mechanism likely helps to prevent unwarranted immune responses against harmless allergens. Supporting this, mice that lack Cul5 in T cells are more susceptible to allergen-induced airways remodeling and asthma.

We found that activating TCR and CD28 receptors on naïve CD4+ T cells increased the expression and neddylation of Cul5. Cul5 levels and activity increased over the first 48 h, aligning with the timing of when CD4+ T cells are making cell fate decisions. TCR and CD28 signals also resulted in a corresponding increase in CIS expression. CIS, via its SH2 domain, would then be available to bind phosphorylated tyrosine residues on client proteins[11,43,44], and recruit these proteins to Cul5. We determined that one of these client proteins is pJak1, a key transmitter of IL-4 receptor signals[21].

Interestingly, Cul5 expression remained elevated in CD4+ T cells even after TCR signals were removed. Thus, while TCR and CD28 signals can initiate Cul5 activity, Cul5 persists in an

activated state. Cul5 levels and activity were roughly similar in CD4[+] T cells that had been differentiated under many different culture conditions, suggesting that Cul5 may function in these cells to prevent them from responding to low levels of IL-4 in the environment, a scenario that would otherwise allow transdifferentiation or loss of lineage commitment.

In activated T cells, Cul5 primarily associated with CIS, but bound to a lesser extent with SOCS2. This was true when T cells were activated in the presence or absence of IL-4. This is interesting because the mammalian genome encodes for ~40 SOCS box proteins, including SOCS family proteins, all of which have the potential to interact with Cul5[13,23], albeit with varying affinities. However, it is not known which, if any, depends on Cul5 for their function. Whether the preference for CIS reflects stronger binding between CIS and Cul5, or that CIS is more abundant than other SOCS family members in these cells, is unclear. However, the physiologic importance of this interaction is reflected by the similarity of phenotypes between *CIS*-deficient mice and *Cul5*[fl/fl]*CD4-Cre* mice[20], as well as CIS-deficient and Cul5-deficient T cells.

SOCS family proteins inhibit cytokine signaling, but how they accomplish this is less clear. SOCS proteins function through three possible mechanisms, binding to pTry residues via their SH2 domain and blocking recruitment of downstream signal transducers, inhibiting kinase activity, or working within a Cul5 complex[11,12,23]. SOCS1 and SOCS3, the most well studied family members, carry out some of their inhibitory functions even in the absence of their SOCS box domain[13]. Furthermore, SOCS family members bind Cul5 with widely varying affinities[23], supporting that some family members may be more dependent on Cul5 than others. CIS has been proposed to act by using its SH2 domain to block recruitment of signal transducers[20], although this has not been shown experimentally. What has been shown is that in the absence of CIS, pSTAT6 levels are elevated. Our data support that CIS regulates pSTAT6 levels by acting as a substrate receptor within a CRL5 complex. Supporting this, CIS was found to associate not only with Cul5, but also other members of the CLR5 complex, including EloB and EloC, CAND1 and CNS proteins. CIS recruits pJAK1 to CRL5 to promote its ubiquitylation and degradation, thus limiting IL-4R signaling.

In the absence of Cul5, CD4[+] T cells became overly sensitive to IL-4 which contributed to asthma pathogenesis. Increased IL-4 sensitivity predisposed CD4[+] T cells to becoming Th2 or, in the presence of TGF-β, Th9 cells. Th2 cells produced IL-4, creating a snowballing effect. Cul5 deficient cells were thus exposed to more IL-4, expanding Th2 and/or Th9 differentiation, and thus exacerbating inflammation and worsening pathology. This scenario most likely explains why 30–36-week-old Cul5 deficient mice developed pronounced Th2 inflammation compared to 8–10-week-old mice which showed little to no Th2 inflammation. Secondly, increased IL-4 sensitivity alters the differentiation of naïve CD4[+] T cells where instead of becoming tolerogenic Treg cells, Cul5 deficient cells are biased to become pathogenic Th9 cells.

The logical question then becomes, how is Cul5 specific for IL-4R signaling and not broadly regulating other cytokine receptors that rely on JAK1? Like CIS, SOCS3 also inhibits JAK1 but only when JAK1 is activated by IL-6 receptor signaling[45]. This specificity is enforced by the binding of SOCS3 to the gp130 receptor. This localisation allows SOCS3 to gain proximity to JAK1[45]. A similar mechanism may be occurring in this context. In support of this we found that CIS associated not only with JAK1 but also with the IL-4Rα. This suggests a model in which CIS binds pIL-4Rα via its SH2 domain and brings the CRL5 complex into proximity with pJAK1 resulting in the ubiquitination and degradation of pJak1. Alternatively, CIS and CRL5 might also directly target IL-4Rα. These

models are supported by other studies showing that CIS binds to phosphorylated receptors[13,43,44]. In either case we believe that the specificity of CRL5 is enforced by recruitment to specific receptors.

While it was known previously that CIS prevented lung remodeling and development of an asthma-like disease by limiting IL-4R signaling[20], the physiologic context for this remained unclear. Based on analyses of CIS transcript abundance, it was proposed that CIS expression required IL-4R signaling. This led to the idea that CIS was upregulated downstream of IL-4R signaling and acted in a negative feedback loop. Our study supports that Cul5 and CIS function downstream of TCR and CD28 activation to set a threshold for IL-4R signaling and thus influence fate choices in differentiating CD4[+] T cells. While our data support that this impacts Treg, Th2, and Th9 differentiation, we did not see differences in Th1 or Th17 differentiation. However, given that we added anti-IL-4 to the Th1 and Th17 cultures, it remains possible that Cul5 might regulate the differentiation of other Th cells. However, our data clearly support that in the absence of Cul5, activated T cells are more sensitive to low levels of IL-4. This has major implications upon allergen exposure. In this scenario, increased IL-4 promotes Th9 cell formation at the expense of Treg cell differentiation, resulting in severe lung remodeling and decreased lung function. Thus, by altering cytokine sensitivity during their peripheral education, T cells can rewire fate choices and set the stage for tissue inflammation and disease development. While we have focused here on these early fate choices, it will be important to know whether Cul5 might continue to impact the long-term plasticity of T cells.

To date, no genetic polymorphism studies have implicated Cul5 or CIS in human allergic diseases such as asthma, although it has been implicated in autoimmune and infectious disease[46]. However, two previous studies support a function for Cul5 and CIS in allergic disease. One study found an association between hsa-miRNA-19b 3p and asthma remission[47]. Using predictive tools, this same study suggested that hsa-miRNA-19b-3p targets Cul5 and CIS. A second study reported that CIS regulates eosinophilic inflammation in bronchial asthma by limiting IL-13 signaling[48]. These studies highlight that CIS and Cul5 might have an important function in asthma, and more generally, in immune-mediated diseases. However, additional studies are needed to determine how Cul5 and/or CIS regulate human T cells to prevent allergic disease and whether these factors might be targeted for therapeutic effect.

## Methods

**Mice.** CD45.1[+], CD45.2[+], Rag knockout (KO), *CD4-Cre* mice were purchased from Jackson laboratories. *Cul5*[fl/fl] mice on a B57BL/6 background were purchased from Taconic. All mice were bred in house at the Children's Hospital of Philadelphia (CHOP). *Cul5*[fl/fl] cage-matched littermates (without *CD4-Cre*) were used as wild type controls except in experiments that required CD45.1[+] control. Mice were used between 8–10 and 30–36 weeks of age. For all the experiments mice were age and gender-matched.

**House Dust Mite treatment and lung function assay.** For House Dust Mite (Greer, catalogue number RMB84M) treatments, mice were anesthetized with 50 mg/kg of ketamine and 5 mg/kg xylazine via intraperitoneal injection. PBS or 20 ug HDM, was placed dropwise on the nose until it was inhaled. Mice were treated once per day for 3 consecutive days on days 0–2, then again for 5 consecutive days on days 8–12. They were analysed on days 13 and 14.

For lung function studies, allergen sensitized and challenged mice were anesthetized with 17.5% ketamine/2.5% xylazine and then cannulated and connected to Buxco FinePointe resistance/compliance computerized system and mechanically ventilated. Lung function was monitored upon methacholine inhalation for 20–30 min in fully anesthetized mice.

**Serum cytokines and ELISA.** Serum was collected from mice and stored at −80ºC. Q-Plex Mouse Cytokine stripwells (16 plex, Quansysbio, catalogue number 110349MS) was used to quantify cytokines upon House Dust Mite (HDM)

exposure according to manufacturer's directions. Total serum antibody ELISA was performed by coating ELISA plates with anti-mouse Ig overnight (Southern Biotech). Plates were blocked, and then standards (Southern Biotech) and serum sample dilutions (1:5000 to 1:125,000) were added and plates were incubated for 2hrs at room temperature. Plates were then washed and isotype-specific detection antibodies conjugated to horseradish peroxidase (Southern Biotech) were added for 1 h at room temperature. The 3,3′,5,5′-tetramethylbenzidine (TMB) substrate was added, and the reaction was stopped by adding 1 M phosphoric acid. Absorbance was measured at at 450 nm.

**Tissue processing and staining.** Spleen, lymph nodes, and lungs were harvested from mice after euthanasia by carbon dioxide and subsequent cervical dislocation. Lungs were treated with collagenase I and Ia (Sigma, catalogue number C0130 and C9891, respectively), and 20 μg/ml DNAse I (Roche, catalogue number 10104159001) in 10 ml plain RPMI for 1 h at room temperature prior to homogenization. Homogenates were treated with 2 ml of RBC lysis buffer for 2 min and passed through an additional 70 μm filter. For intracellular staining, $5*10^6$ cells from these homogenates were resuspended in 1 ml of RPMI 1640 supplemented with 10% FCS, 100U/ml Penicillin-streptomycin, 1X Glutamax (ThermoFisher, cat. 35050061), 1X Non-essential amino acids (Gibco, cat. 11140-050), 2 mM HEPES (Gibco, cat. 15630-080), 1 mM sodium pyruvate (Corning, cat. 20115013), 8 μl/L 2-mercaptoethanol (Sigma), 1 μg/ml Golgi Plug (BD, cat. 555029), 1 μg/ml Golgi Stop (BD, cat. 554724), PMA (30 ng/ml, Calbiochem) and Ionomycin (1 μM, Abcam). These resuspensions were incubated at 37 °C for 4 h before proceeding with staining.

Single cell suspensions were first stained with a fixable viability dye (Life Technologies, cat. L34961, 1:60) and pre-treated with unlabeled anti-CD16/CD32 (Fc Block, BD Pharminogen, cat. 564219, 1:1000). Cells were surface stained for 15 min at 4 °C in FACS buffer with the following antibodies: anti-CD4 (Biolegend, clone GK1.5, 1:200), anti-CD3 (Biolegend or BD biosciences, clone 17A2, 1:100), anti-CD62L (Biolegend, clone MEL-14, 1:100), anti-CD44 (Biolegend, clone IM7, 1:200), anti-CD45.1 (Biolegend, clone A20,1:200), and anti-CD45.2 (Biolegend, clone 104, 1:200), anti-CD124 (BD, mIL4R-M1, 1:50). After staining, cells were then washed with FACS buffer and fixed in 50 μl BD Cytofix/Cytoperm (BD, cat 554714) for 20 min in ice. For intracellular cytokine staining cells were washed in 100 μl BD perm wash buffer and stained for intracellular cytokine in BD perm wash buffer (cat. 554723) with following antibodies: anti-IFNγ (BD, clone XMG1.2, 1:100), anti-IL2 (Biolegend, clone JES6-5H4, 1:100), anti-IL5 (BD, clone TRFK5, 1:100), anti-IL4 (BD, clone 11B11, 1:100), anti-IL13 (ebioscience; clone eBio13A, 1:100), anti-IL17A (Biolegend, clone TC11-18H10.1, 1:100), antiIL-9 (Biolegend, clone RM9A4, 1:100). For mast cell staining lung homogenate was stained with anti-CD45 (BioLegend, clone 30 F-11, 1:200), anti-FcεRI (ebiosciences, clone Mar-1, 1:100), and anti-CD117 (Biolegend, 2-B8, 1:100) for 15 min at 4 °C and fixed using BD cytofix/cytoperm buffer. For eosinophils lung homogenate was stained with anti-CD45 (Biolegend, clone 30 F-11, 1:200), anti-Siglec F (BD, clone E50-2440, 1:100), anti-CD11c (Biolegend, clone N418, 1:200) and anti-CD11b (Biolegend, clone M1/70, 1:200) for 15 min at 4 °C and fixed using BD cytofix/cytoperm buffer.

**Cell isolation and culture.** Spleen and lymph nodes were harvested from mice as described above. Single cell suspensions were resuspended in MACS buffer consisting of 0.5% fetal calf serum (FCS; Atlanta Biologicals Premium) and 2 mM EDTA in PBS. Naïve CD4+ T cells were isolated using a naïve T cell isolation kit (Miltenyi, cat. 130-104-453) following manufacturer's directions. This method resulted in >90% CD4+CD62L+CD44 CD25- cells.

Cells were cultured in RPMI 1640 supplemented with 10% FCS, 100 U/ml Penicillin-streptomycin, 1X Glutamax (ThermoFisher, cat. 35050061), 1X Non-essential amino acids (Gibco, cat. 11140-050), 2 mM HEPES (Gibco, cat. 15630-080), 1 mM sodium pyruvate (Corning, cat. 20115013), 8 ul/L 2-mercaptoethanol (Sigma). Cells were cultured at 37 °C in 10% carbon dioxide. To activate T cells, cell culture plates were coated using 5 μg/ml anti-CD3 (BD biosciences, clone 17A2) and 5 μg/ml anti-CD28 (BD biosciences, clone 37.51) in PBS at 4 °C overnight or at least 2 h at 37 °C. For restimulation experiments, cells were stimulated with anti-CD3 anti-CD28 dynabeads (Gibco, cat. 20115013) at a 1:3 beads:cell ratio. For CHX experiments cells were stimulated with 5 μg/ml anti-CD3 and 2 μg/ml anti-CD28 for 48 h. Cells were then washed with RPMI and stimulated with IL-4 for 2 h. After treatment cells were washed and treated with CHX for the indicated time. For Immunoprecipitations experiments, approximately 50–60 million cells were stimulated with 5 μg/ml anti-CD3 and 2 μg/ml anti-CD28 for 4hrs in presence of 10 ng/ml IL-4 or anti-IL-4 (10 ng/ml) along with NAEi (MLN4924 1 μM; Lifesensor).

**In vitro differentiation.** Naive CD4+CD25−CD62L$^{hi}$CD44$^{lo}$ T cells were isolated using Naïve isolation kit (Miltenyi catalogue number 130-104-453) and activated with 5 μg/ml plate-bound anti-CD3 (Biolegend) and 5 μg/ml soluble anti-CD28 (Biolegend) in the presence of polarizing cytokines or/and blocking antibodies. For T$_H$1 polarization cells were cultured in presence of 20 μg/ml anti-IL-4 (11B11; BioXCell), 20 ng/ml IL-12 (Peprotech; 210-12), and 10 U/ml IL-2 (Peprotech; 200-02) for 5 days. For T$_H$17 polarization, cells were cultured in media containing

20 ng/ml IL-6 (Peprotech; 500-P56) and 0.5 ng/ml TGF-β (Peprotech; 100-P21), 20 ng/ml IL-1β (Peprotech; 500-P51), 50 ng/ml IL-23 (Peprotech; 200-23) and 2 μg/ml anti-IL-2 (Biolegend; JES6-1A12) for 5 days. For T$_H$2 polarization anti-CD3/CD28 (GIBCO; 11452D) dynabeads were used at 1:1 (Bead:cell) ratio. Naïve CD4+CD25−CD62L$^{hi}$CD44$^{lo}$ T cells were cultured in media with 20 μg/ml anti-IFNγ (Biolegend, XMG1.2), 10 ng/ml IL-4 (Peprotech; 500-P54), 20 μg/ml anti-IL-12 (Biolegend; C18.2) and 50 U/ml IL-2 (Peprotech; 200-02). For T$_H$9 polarization 2 μg/ml plate bound anti-CD3 and 1 μg/ml soluble anti-CD28 antibody was used. Naïve CD4+CD25−CD62L$^{hi}$CD44$^{lo}$ T cells were cultured in media supplemented with 10 μg/ml anti-IFN-γ (Biolegend, XMG1.2), 2.5 ng/ml TGF-β (Peprotech; 100-P21) and 20 ng/ml IL-4 (Biolegend; 500-P54). Alternately, to check the differentiation fate, naïve CD4+CD25−CD62L$^{hi}$CD44$^{lo}$ T cells were cultured in 96 well plates coated with 2 μg/ml anti-CD3 and soluble 1 μg/ml anti-CD28 in Treg conditioning media (0.5 ng/ml TGF-β and 50 U/ml IL-2 along with 10 μg/ml anti-IFNγ) in presence of increasing concentration of IL-4 (0–10 ng/ml).

**Mixed Chimeras.** Bone marrows cells were harvested from *Cul5*$^{fl/fl}$ *CD4-Cre* CD45.2+ mice and wild type CD45.1+ mice and frozen at −80 until use. Upon thawing, bone marrow cells were T cell depleted using TCRβ PE antibody and anti-PE microbeads (Miltenyi, catalogue number 130-048-801). Rag KO recipients were irradiated with 400 Rad using an X-Rad Irradiator, and subsequently injected via tail vein with $2 × 10^6$ bone marrow cells in RPMI 1640 (Hyclone, SH30027.01) with 100 U/ml Penicillin-streptomycin (Gibco, cat. 15140122) at a 1:1 mixture of wild type: *Cul5*$^{fl/fl}$ *CD4-Cre*. Mice were maintained on antibiotic (trimethoprim-sulfamethoxazole) containing water for 2 weeks following irradiation and injection and allowed to reconstitute for at an additional 6 weeks. They were then treated with HDM as described above.

**Immunoprecipitation.** Cells were stimulated with either plate bound 5 μg/ml anti-CD3 and soluble 2 μg/ml anti-CD28 or by anti-CD3/CD28 dynabeads (1 bead:3 cell) in presence of 10 ng/ml IL-4 (Peprotech) or 20 μg/ml anti-IL-4 or 10 ng/ml IL-4 along with 1 μM NAEi (Lifesensors; S19830) for 4 h. Cells were then lysed in NP-40 buffer (50 mM Tris-cl, pH 7.4, 1 mM EDTA, 100 mM NaCl, 1% NP-40) in the presence of deubiquitylase inhibitor PR619 (Lifesensors; S19619), 1,10 phenanthroline (Lifesensors; S19649), Protease inhibitor cocktail EDTA free (Roche) and phosphatase inhibitor cocktail (Thermo). After 30 min of lysis, the samples were centrifuged at 15,000 rpm for 10 min at 4 °C. Cleared supernatant was quantified using Bradford reagent and 3–4 mg protein was used for Immunoprecipitation. For preclearing, supernatants were incubated with 2 μg anti-IgG conjugated with protein A dynabeads for 2 h at room temperature. Precleared supernatant was then incubated with either anti-Cul5 (2 μg), or anti-CIS (3 μg) or anti-IgG (2 or 3 μg) conjugated with protein A dynabeads for overnight at 4 °C. Beads were then washed five times with 1X TBS containing 0.1% Tween 20. The immunoprecipitated proteins were eluted with 2X Laemmli buffer (240 mM Tris, pH 6.8, 8% SDS, 0.04% bromophenol blue, 5% β-mercaptoethanol, 40% glycerol) and boiled at 95 °C for 5 min.

**Mass spectrometry.** Samples were analyzed on a QExactive HF mass spectrometer (Thermofisher Scientific San Jose, CA) coupled with an Ultimate 3000 nano UPLC system and and EasySpray source. Peptides were separated by reverse phase (RP)-HPLC on Easy-Spray RSLC C18 2 um 75 μm id × 50 cm column at 50 C. Mobile phase A consisted of 0.1% formic acid and mobile phase B of 0.1% formic acid/acetonitrile. Peptides were eluted into the mass spectrometer at 300 nL/min with each RP-LC run comprising a 90 minute gradient from 1 to 5% B in 15 min, 5–45% B in 90 min. The mass spectrometer was set to repetitively scan m/z from 300 to 1800 ($R = 120,000$), followed by data-dependent MS/MS scans ($R = 45,000$) on the twenty most abundant ions, normalized collision energy (NCE) of 27, dynamic exclusion of 15 s with a repeat count of 1. FTMS full scan AGC target value was 5e5, while MSn AGC was 1e5, respectively. MS and MSn injection time was 120 ms; microscans were set at one. Unassigned, 1, 7, 8, and >8 charge states were excluded from fragmentation. Mass spectrometry data is submitted in proteome exchange. The project Accession is PXD028492.

**Western blotting.** After treatment cells were washed with phosphate-buffered saline (PBS) and lysed using 1% Triton-X lysis buffer containing 50 mM Tris, 100 mM NaCl, 1X Protease inhibitor cocktail EDTA free (Roche), 1X Halt phosphatase inhibitor cocktail (Thermo Fisher Scientific), Zn$^{2+}$ chelator *ortho*-phenanthroline (o-PA) (LifeSensors) and deubiquitylase inhibitor PR-619 (LifeSensors). Protein lysate were quantitated using Bradford reagent. For Cul5 neddylation experiments whole-cell lysates were prepared as described previously[49]. Briefly, cells were resuspended in SDS sample buffer (62.5 mM Tris-HCl, 2% w/v SDS, 10% glycerol, 50 mM DTT, and 0.1% bromophenol blue), vortexed to reduce sample viscosity, denatured by boiling, and then cooled on ice. This is to prevent the de-neddylation during protein extractions. Samples were resolved on Novex Tris-Glycine Gels (Thermo Fisher Scientific) and then transferred onto PVDF membrane (Amersham Pharmacia Biotech, Piscataway, NJ). The membrane was probed with the primary antibodies Cul5 (Bethyl Lab; 1:5000), pJak1 (CST; 1:1000), Jak1 (BD; 1:1000), pSTAT6 (CST; 1:1000), STAT6 (CST; 1:1000), IL4Rα (BD; 1:1000), pIL4Rα (BD; 1:1000), Actin (SCBT; 1:5000), Cul5 (Bethyl; 1:5000). The

membrane was then incubated with an appropriate secondary antibody at a dilution of 1:5000 and the protein of interest was detected with Li-Cor odyssey imaging system.

**RNA-seq.** Naïve CD4$^+$ T cells were isolated by MACS from WT and *Cul5*$^{fl/fl}$*CD4-Cre* mice. Cells were cultured under Th9 polarising conditions for 48 h. Total RNA was isolated using TRIzol reagent (Thermo Fisher Scientific) and poly-A selection was used to remove ribosomal RNA. After first and second-strand synthesis from a template of poly-A selected/fragmented RNA, other procedures from end-repair to PCR amplification were performed. The DNA library was quantitated using Qubit. Libraries were sequenced on BGIseq500 platform with 50 bp single-end sequencing. Single-end sequencing reads were quality checked using FASTQC (www.bioinformatics.babraham.ac.uk/projects/fastqc) and reads were pseudo-aligned to the Ensembl Mus musculus reference genome (GRCm38) using Kallisto (V0.46.2). The transcript abundances were assembled and TMM-normalized log2CPM (count per million fragments) was estimated using the Bioconductor package edgeR (V3.34.0) for differential expression analysis. RNAseq data has been submitted in Gene Expression Omnibus (GEO). The accession number is GSE184321.

**Public RNA-Seq datasets.** For Fig. 5, heatmaps were generated by analysing public datasets retrieving Naïve CD4$^+$, Th0, Th1, Th2, Th9, and Th17 raw RNAseq data from **GSE153862** and aligned and quantified (reads per kilobase per million reads [RPKM]) using Kallisto. The fold-change for each T helper subset were calculated with respect to the naïve CD4$^+$ quantified RPKM. For each T helper subset, the genes with log2FC greater than 5 were used for generation of the circle plot, and of those, the genes corresponding to the top 200 log2FC were taken as the "top hit gene sets" for the respective T helper subset to be used for the heatmaps. For the Cul5-KO RNAseq data, log2FC were calculated as log2 of KO over WT (units of RPKM).

Microarray data for 6 h re-stimulated and unstimulated Th9-differentiated cells were retrieved from **GSE44937** using the GEOparse package in Python. The log2 fold-changes and student's *t*-test *p*-values of stimulated over unstimulated subsets were used to obtain a list of "top hit genes" where the log2FC > 2 and the *p*-value <0.05.

Heatmaps were created using Morpheus software (Broad Institute) by taking the top 200 hits gene set for each T helper subset and comparing the overlap with the differentially expressed Cul5-KO RNA-seq dataset (a Student's *t*-test was used with *p*-value <0.05 as the cut-off).

The genes from the public datasets above with log2FC greater than 5 were used to create a circle plot using the GOCircle function of the GOPlot package in R. Z-scores were calculated using Eq. (1):

$$zscore = \frac{\#upregulated\ genes - \#downregulated\ genes}{\sqrt{gene\ count}} \quad (1)$$

The inner circle represents -log10(adjusted *p*-values)—the larger the section, the more significant the finding. Adjusted *p*-values for the inner circle were calculated using Fisher's Exact Test on whether the gene subset was enriched in the differentially upregulated population within the entire Cul5-cKO RNA-seq dataset and then applying the Benjamini-Hochberg procedure to obtain the adjusted *p*-value for multiple hypothesis testing.

To compute the binding sites STAT6 and STAT5B ChIP-seq were retrieved from **GSE41317**. BATF, PU.1, and IRF4 ChIP-seq were retrieved from **GSE99165**. Raw reads were uploaded into GalaxyProject and aligned with Bowtie2 using the mm10 genome build as the reference, producing paired-end BAM files. Aligned reads were filtered with MAPQ > 20 and split into sample read-groups for peak calling. Peak calling was achieved with the MACS2 call peaks function using background controls provided by the dataset. For the STAT6 and STAT5B ChIP-seq, an FDR threshold of $q < 1e-20$ was used due to a large number of binding sites requiring additional filtering. For the other ChIP-seq sets, an FDR threshold of $q < 0.01$ was used. This led to gene set sizes of 767, 1552, 674, 494, and 27 genes for STAT6, STAT5B, BATF, PU.1, and IRF4, respectively.

**Histology scoring.** H&E scoring was performed using Aperio Image Scope software. We counted the total number of nuclei as a surrogate for cell number using the Nuclear V9 setting. The total number of nuclei was divided by the total area to get the number of nuclei/mm$^2$. These values were multiplied by 100 to get the number of nuclei/100 mm$^2$ area of lung. Goblet cell hyperplasia was scored based on previously described methods[50,51]. Briefly, five airways were included for scoring based on following parameters, 0: No visible hyperplasia or mucous production, 1: patchy hyperplasia and/or PAS staining in 25%, 2: patchy hyperplasia and/or PAS staining in <50% of bronchioles, 3: patchy hyperplasia and/or PAS staining in <75%, 4: criteria for 3 plus bronchiolar plugging or obliteration. Scores reported were the total score for each lung (0–15).

**Statistics.** Results were expressed as the mean ± standard error of mean (SEM) or mean+standard deviation (SD). Statistical analysis was performed using Prism software version 7 and 9. *p*-value was calculated using the two-tailed student *t*-test.

**Reporting summary.** Further information on research design is available in the Nature Research Reporting Summary linked to this article.

## Data availability

The RNAseq raw data are deposited and available from the GEO database under the accession number GSE184321. IP-MS/MS data from D10 cells are deposited and available from PRIDE database under accession number PXD028492. Other datasets used in this study are from Bulk RNA-seq of mouse Th cell subtypes GSE153862, microarray data of mouse Th9 cells GSE44937, mouse ChIP-seq data GSE41317 and GSE99165. All other data supporting the findings are available in this paper or from the corresponding author upon request. Source data are provided with this paper.

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

## Acknowledgements

We thank the Proteomics, Flow Cytometry and Pathology Core at Children's Hospital of Philadelphia. This work was supported by the National Institutes of Health Grants (R01AI148240) to P.M.O.

## Author contributions

B.K., N.F., and P. O. designed the experiments and drafted the manuscript. B.K. performed the majority of the experiments and assembled the figures. Y.C., A.S., E.M., A.D., D.K., K.S., N.P., P.S., and O.E. provided additional technical and/or intellectual expertise. D.R.H. provided guidance on the HDM experiments. C.F.P. and L.Y.H. provided guidance and assisted with HDM and airway resistance experiments. All authors read/edited the manuscript.

## Competing interests

The authors declare no competing interests.

## Ethics

All procedures were conducted in accordance with the Animal Welfare Act and were approved by the Institutional Animal Care and Use Committee at Children's Hospital of Philadelphia (protocol ID 810).
