## [Peer Review File · Nature Communications]

Cul5 regulates CD4+ T cell fate choice and allergic inflammationREVIEWER COMMENTS

Reviewer #1 (Remarks to the Author):

Cul5 regulates CD4+ T cell fate choice and allergic inflammation 2 by Binod Kumar*

Major points:

The manuscript fails to report any evidence of the clinical role of Cul5 in human especially with regard to allergies and for this and other missing informations in the allergy model, the conclusions and the title are not scientifically documented.

The authors discovered a spontaneous phenotype of lung inflammation in Cul5fl/fl 242 CD4-Cre mice during aging (Fig 2). Unfortunately the authors did not follow up the reason why these mice develop this phenotype after aging and moved to a model of allergic asthma (Fig 3).

Regarding the inflammatory score and other histopathological read out for Figure 2 and 3 quantifications are missing. Especially for Figure 3 the eosinophilia looks not high as described in asthma models (60-70%). In figure 3 also differential cell count in the BALF should be shown. For figure 3 important cytokines are missing like IL-13 and TGF-beta especially regarding the remodeling present in allergic asthma. The conclusions from different key figures are made based on 5 mice which is too low (see IL9+ T cells and mast cells).

In figure 3 and following figures using CD4 + T cells from Cul5fl/fl 242 CD4-Cre and control Cul5fl/fl mice it is not clear how old the mice were . Also here the authors should have compared the model of asthma in young versus old mice to make their original observation more interesting. Also the authors should have shown the IgE results in the asthma model and the classical Th2 and Th1 staining in this model .

Overall the asthma model was not fully investigated and the authors moved to naïve cells investigations.

Next by describing Figure 4 the authors add to many literature informations that probably belong to the introduction. In figure 4 the authors should have used CD4 + T cells from Cul5fl/fl 242 CD4-Cre and control Cul5fl/fl mice , instead of using antibodies and IGG controls.

In this figure the number of observation is equal 1 which makes uncertain the reader although one could appreciate the number of methods used.

In figure 5 the authors analyzed 3 mice per group.

Did the authors identified Cul5 downregulated in the wild type under Th9 and Th2 skewing conditions in the top hit genes? Did the authors analyzed RNA seq under T reg skewing conditions? Figure 6 has the same problem as shown for figure 4. Why did not the authors use CD4 + T cells from Cul5fl/fl 242 CD4-Cre and control Cul5fl/fl mice , instead of using antibodies and IGG controls in the immunoprecipitation assay? Again the n per group is also low.

Figure 7 . The author should look for Th2 upregulation like CD4+GATA3+ T cells upon IL-4 increase.

Overall the explanation of the authors of an interconversion between Th9 and T reg mediated via Cul5 would require additional experiments as it looks like the authors have a mixture of TH9,Th2 and T regulatory cells in the experiments shown in Figure 7a, b.

Reviewer #2 (Remarks to the Author):

In this manuscript, Kumar et al describe the phenotype of T cell conditional Cul5 KO mice. They described that Cul5 deficient mice have a Th2 and Th9 bias and the mice under physiological conditions develop allergic airway inflammation. They propose that the mechanism is by enhancing IL-4R signaling by stabilizing Jak1. At the end of the manuscript, the author also describe that Cul5 deficient T cells deviate from Treg to Th9 cells. Overall, the study provides a large number of observations and potential mechanisms but none is fully demonstrated and at the end of the manuscript it is not clear what the message is. The major interesting observation is that Cul5 deficient mice develop specifically allergic airway inflammation under physiological conditions. How

this happens is not addressed in the study. It is unclear whether this is a developmental problem, which type of antigen is activating this inflammatory reaction, or even whether this is selectively mediated by CD4 cells. The section at the end of the manuscript regarding Treg cells is even more confusing. If the authors eventually think that it is a reduced number of Treg that cause this inflammatory response in the lung, they have to test this hypothesis. The manuscript is poorly written and it is difficult to follow the rationale. Their data in different figures often contradict each other.

1. From Fig. 1A, it is concluded that the levels of unneddylated Cul5 decreased upon activation, while the neddylated fraction increases. However, the results in Fig. 1B do not support these conclusions since the levels of unneddylated are the same or higher. There is inconsistency in the results among figures. This raises questions regarding the reproducibility of the data and conclusions.

2. Since little is known about the CD4-Cul5 conditional mice, the authors need to examine the role in T cell development in these mice.

3. In Fig. 2, they use CD44 as a marker for effector cells, but this is not correct. CD44 is high in memory cells as well, and it is a marker for homeostatic proliferation. Thus, it cannot be concluded that Cul5 conditional mice have higher frequency of effector CD4 cells. In addition, in this figure (lung infiltration under physiological conditions) the authors need to examine the presence of other immune cells, not just CD4, including CD8, macrophages, eosinophils, neutrophils, etc.

4. In Fig. 3, they examine the allergic airway inflammation in response to exposure to HDM. Of course, it is not unexpected that the Cul5 conditional mice have increased allergic airway inflammation in response to HDM since they already have allergic airway inflammation in basal conditions. The key question is what is triggering the allergic airway inflammation in the absence of any insult. This is what is missing in this study, are CD4 cells autoreactive? are these generated during development?

5. There is more inconsistency among the results in Fig. 2 and Fig. 3. Fig. 2 shows increased frequency of CD4 cells producing IL-4, but Fig. 3 in PBS mice (should be equivalent to mice in Fig. 2) shows no difference in the production of IL-4.

6. Regarding IL-9 production, again the major difference is in WT and Cul5 under physiological conditions, being higher in CD4 cells from Cul5. The authors need to show the presence of mast cells in the mice prior to exposure to HDM.

7. While it is concluded that "Cul5 limits the frequency and number of Th9 cells and prevents lung remodeling and other pathologic features associated with asthma" the phenotype is not restricted to Th9 cells, since Th2 cytokines are also affected. The essential question is how Cul5 leads to accumulation of these cells in the absence of antigen.

8. Most of the biochemistry studies in Fig. 4 were performed with D10 cells, but D10 cells are far from representing either Th2 or Th9. These studies must be repeated with primary Th2 cells and Th9 cells.

9. In Figure 4, from mouse Th0 and Th9 cells they conclude that CIS upregulation is not mediated by IL-4, but they do not present data with CD4 cells under Th2 conditions. Furthermore, the results in human CD4 cells contradict their own conclusions since anti-IL-4 Ab reduce CIS levels. Thus, at the end, the reviewer (and most likely potential readers) has no clarity regarding the regulation of CIS in primary CD4 cells.

10. The authors propose that Cul5 deficiency predominantly favors Th9 differentiation. The data in the study do not support this conclusion. Based on the results from Fig. 5, all Th2, Th1 and Th9 genes were clearly upregulated in Cul5 deficient mice.

11. If the mechanism for Cul5 to regulate Th2 differentiation is through Jak1 degradation, the response to many other cytokines using Jak1 should also be affected. The authors have to

examine the signaling and response to more cytokines that use Jak1.

12. There are several issues concerning the written manuscript. Poorly. written. For instance, it was difficult to find Figure legends and this is because they were placed before the Methods section. "IP'ed" does not exist as a word

Reviewer #3 (Remarks to the Author):

This is an interesting manuscript describing a novel role for Cul5. The authors propose that Cul5 regulates IL-4 receptor signaling and thereby controls the fate of Th cells. They show in ko and molecular interaction assays that mice lacking Cul5 have the propensity to develop Th2 and Th9 driven inflammation and pathology.

The manuscript is very well written and the experiments are conclusive.

Comments:

1. Page 8, line 172: Please change "unprovoked" to "spontaneous" in the title of this chapter and throughout the manuscript.

2. Page 8, lines 183-190: The explanation and logic provided in this and the following sentences that goblet cell hyperplasia is caused by IL-4 and IL-5 is wrong. Instead, IL-13 is the major driver for goblet cell hyperplasia, however, IL-13 data are not shown here.

The reviewer suggests to show IL-13 data along with the IL-4 and IL-5 data. This would be much more convincing and would follow the line of arguments regarding increased numbers of goblet cells.

3. Page 9, line 204: Please change phrasing to "...we challenged mice with house dust mite extract (HDM)...", because it is the extract made of mites with which you challenge.

Figure 1F: Please change the graphs for lungs to representative ones. Fig 1G shows what is representative, however, in Fig 1F two extreme examples of what has been found in mice is depicted.

Figure 2. Change title from "unprovoked" to "spontaneous".

Figure 2: What is somewhat disturbing in this figure is that fact that the graphs from E-J have dissimilar numbers of animals. This is true not only for the determination of different data, e.g., percent positive cells and absolute numbers of IL-4 and IL-5 T cells and mouse immunoglobulin levels for IgE and IgG1 but also within the experiments showing IgE versus IgG1 levels. This should be really avoided.

Figure 3C: Here one wonders, why no IL-9 data are presented. Later in the ductus of the manuscript IL-9 gets a very important place. It would be interesting to see whether or not IL-9 levels are also elevated in BAL.

Figure 5D-F and corresponding Results-section page 14 (lines 329-338): This part is difficult to understand. Especially the point that "fewer genes were identified for Th2, Th1, Th17 and Th0 cells". What exactly is meant by this, given the fact that, e.g., the Top Th2 hits shared many of the upregulated genes also seen in the Th9 Top hit genes?

We appreciate the Reviewers' consideration of our manuscript. While all three reviewers found the data interesting and important, the reviewers asked us to clarify the reason *Cul5^{fl/fl}*CD4-Cre mice develop spontaneous allergy and expand our studies in the asthma model. In this revised manuscript, we have attempted to provide additional data to address these concerns. Our point-by-point response to the reviewers is as follows.

All the changes or points that address reviewers' comments are highlighted in the manuscript in yellow.

Reviewer #1 (Remarks to the Author):

Major points:

1. The manuscript fails to report any evidence of the clinical role of *Cul5* in human especially with regard to allergies and for this and other missing information's in the allergy model, the conclusions and the title are not scientifically documented.

We have now detailed what is currently known regarding the role of *Cul5* and *CIS* in human asthma. This is in the discussion and is copied below.

"These observations were mostly performed in mice or using primary mouse cells, however, key findings were validated in human CD4+ T cells. These data support that *Cul5* and *CIS* might also regulate human T cell biology and asthma susceptibility. To date, no genetic polymorphism studies have implicated *Cul5* or *CIS* in asthma, although it has been implicated in autoimmune and infectious disease (PMID:20484391). However, two recent studies support a role for *Cul5* and *CIS* in allergic disease. One study found an association between hsa-miRNA-19b 3p and asthma remission (28238746). Using predictive tools, this same study suggested that hsa-miRNA-19b-3p targets *Cul5* and *CIS*. A second study reported that *CIS* regulates eosinophilic inflammation in bronchial asthma by limiting IL-13 signaling (PMID:30197185). These studies highlight that *CIS* and *Cul5* might play an important role in asthma, and more generally, in immune-mediated diseases. However, additional studies are needed to determine how *Cul5* and/or *CIS* regulate human T cells to prevent allergic disease and whether these factors might be targeted for therapeutic effect"

Based on reviewers comments we have also further characterized the allergy model. Specific comments regarding these additions are detailed below.

2. The authors discovered a spontaneous phenotype of lung inflammation in *Cul5^{fl/fl}* 242 CD4-Cre mice during aging (Fig 2). Unfortunately, the authors did not follow up the reason why these mice develop this phenotype after aging and moved to a model of allergic asthma (Fig 3).

In the revised version of manuscript, we have clarified the reason why 30–36-week-old mice developed Th2 inflammation. The section detailing this is found in the discussion section and is copied below:

“In the absence of Cul5, CD4+ T cells became overly sensitive to IL-4 which contributed to asthma pathogenesis. Increased IL-4 sensitivity predisposed CD4 T cells to becoming Th2 or, in the presence of TGF- β , Th9 cells. Th2 cells produce IL-4 creating a snowballing effect. Cul5 deficient cells are thus exposed to more IL-4, expanding Th2 and/or Th9 differentiation, and thus exacerbating inflammation and worsening pathology. This scenario most likely explains why 30–36-week-old Cul5 deficient mice developed pronounced Th2 inflammation compared to 8–10-week-old mice which showed little to no Th2 inflammation.”

3. Regarding the inflammatory score and other histopathological read out for Figure 2 and 3 quantifications are missing. Especially for Figure 3 the eosinophilia looks not high as described in asthma models (60-70%). In figure 3 also differential cell count in the BALF should be shown.

In the revised manuscript we have added the histopathological score. The results are shown in Fig.2c, d and Fig. 3b, c. The trichome staining did not show a significant difference, thus it has been removed from the figure. The methodology for scoring is described in methods section of manuscript.

In this study we observed that eosinophil numbers and frequencies were increased in HDM treated group compared to PBS treated group (Fig. 3d, g, h). Eosinophil numbers in the BAL are similar to two previously published studies using a similar allergy model and treatment regime (Coquet et al. PMID: 26287681; Woo et al. PMID: 29720689). It is known that mouse facilities and gut flora can impact the severity of allergic and autoimmune models between facilities. While we cannot explain why our lung eosinophil numbers might be lower than some other published studies, our results are highly reproducible and reflect numbers in mice that are fully backcrossed onto C57BL/6.

The differential count in BAL has been added and is now shown in Fig. 3d-f and Supplementary Fig. 3c, d.

4. For figure 3 important cytokines are missing like IL-13 and TGF-beta especially regarding the remodeling present in allergic asthma. The conclusions from different key figures are made based on 5 mice which is too low (see IL9+ T cells and mast cells).

Based on reviewers' recommendation we have now provided IL-13 and TGF- β data in the revised version of the manuscript. The results are shown in Fig. 3k, l and Supplementary Fig. 3i.

In addition, we have increased the number of replicates and included a PBS controls for mast cell numbers. Results are shown in Fig. 3o, p. The number of mice for each experimental panel is provided in the figure legend.

5a. In figure 3 and following figures using CD4 + T cells from Cul5fl/fl 242 CD4-Cre and control Cul5fl/fl mice it is not clear how old the mice were. Also, here the authors should have compared the model of asthma in young versus old mice to make their original observation more interesting. Also, the authors should have shown the IgE results in the asthma model and the classical Th2 and Th1 staining in this model. Overall, the asthma model was not fully investigated, and the authors moved to naïve cells investigations.

The age of the mice for each experiment is now clarified in each figure legend.

The purpose of doing HDM experiments in younger mice was that young Cul5fl/fl CD4-Cre mice exhibit very little Th2 inflammation. So, using younger mice allowed us to test the allergic predisposition of Cul5fl/fl CD4-Cre mice and controls at a time when the mice had similar baselines of inflammation. Given that older mice show higher Th2 inflammation, we reasoned that HDM treatment of mice with preexisting inflammation would not be an interpretable comparison.

Based on the reviewers' comments, we have added Th1 data in Supplementary Fig. 3k and HDM specific IgE in Supplementary Fig. 3l.

Next by describing Figure 4 the authors add to many literatures information's that probably belong to the introduction. In figure 4 the authors should have used CD4 + T cells from Cul5fl/fl 242 CD4-Cre and control Cul5fl/fl mice, instead of using antibodies and IGG controls. In this figure the number of observation is equal 1 which makes uncertain the reader although one could appreciate the number of methods used.

We have modified the result section for Figure 4 to reduce the background information about CIS.

We appreciate the reviewer's comment about using Cul5fl/flCD4Cre CD4 T-cells as a control. However, in practice, we have found the using the isotype control allows us to better subtract background. When we enrich for CD4 T cells, a small (but not nonexistent) population of cells contaminates the cultures. Given that these are CD4-Cre mice, the contaminating cells are Cul5 positive and thus in the IP, a small amount of Cul5 is evident in the control MS/MS. This does not happen when we use isotype control. Because we need large numbers of cells needed for these experiments, it is difficult for us to sort to get a very highly pure population of cells. For these reasons, we use isotype control.

To clarify, the results shown in Fig. 4 are not from 1 replicate. They are as follows:

Fig. 4a - The results are from two biologic replicates.

Fig. 4b - The result is representative of two independent experiments.

Fig. 4e - The mass spectrometry results shown are from two biologic replicates.

Fig. 4f - Results are representative of three independent experiments.

This information is now clearly provided in legends of the manuscript.

In figure 5 the authors analyzed 3 mice per group. Did the authors identified Cul5 downregulated in the wild type under Th9 and Th2 skewing conditions in the top hit genes? Did the authors analyzed RNA seq under Treg skewing conditions?

Yes, we have used three mice per group in panels c-h.

In the RNAseq experiment we compared gene expression in wildtype and Cul5KO cells under Th9 skewing conditions only. We did find that Cul5 was significantly lower in the Cul5 deficient cells. We have not performed RNA seq under Treg skewing conditions.

8a. Figure 6 has the same problem as shown for figure 4. Why did not the authors use CD4 + T cells from Cul5fl/fl 242 CD4-Cre and control Cul5fl/fl mice, instead of using antibodies and IGG controls in the immunoprecipitation assay? Again, the n per group is also low.

Please see above for the isotype versus Cul5-deficient cells justification.

To validate the finding that Cul5 and CIS interact with JAK1, we used both Cul5 and CIS as bait proteins in separate experiments. The interaction of CIS with Jak1 was detected by mass-spectrometry (Fig 6c) and the interaction between Cul5 and Jak1 was shown by immunoblot (Fig 6h). These data support that CIS and Jak1 associate within a CRL5 complex.

The detail for replicates is:

Fig. 6c - The results shown are from two biologic replicates.

Fig. 6h - The results shown are representative of two independent experiments.

Figure 7. The author should look for Th2 upregulation like CD4+GATA3+ T cells upon IL-4 increase. Overall the explanation of the authors of an interconversion between Th9 and T reg mediated via Cul5 would require additional experiments as it looks like the authors have a mixture of Th9, Th2 and T regulatory cells in the experiments shown in Figure 7a, b.

We have now included data showing IL-4 staining in these cultures Supplementary Fig. 6a. Given that no IL-4 staining was observed, we did not stain for GATA-3. As has been well documented, GATA-3 is not unique to Th2 cells and GATA-3 staining is not a reliable marker for defining Th2.

Reviewer #2 (Remarks to the Author):

In this manuscript, Kumar et al describe the phenotype of T cell conditional Cul5 KO

mice. They described that Cul5 deficient mice have a Th2 and Th9 bias and the mice under physiological conditions develop allergic airway inflammation. They propose that the mechanism is by enhancing IL-4R signaling by stabilizing Jak1. At the end of the manuscript, the author also describes that Cul5 deficient T cells deviate from Treg to Th9 cells. Overall, the study provides a large number of observations and potential mechanisms but none is fully demonstrated and at the end of the manuscript it is not clear what the message is. The major interesting observation is that Cul5 deficient mice develop specifically allergic airway inflammation under physiological conditions. How this happens is not addressed in the study. It is unclear whether this is a developmental problem, which type of antigen is activating this inflammatory reaction, or even whether this is selectively mediated by CD4 cells. The section at the end of the manuscript regarding Treg cells is even more confusing. If the authors eventually think that it is a reduced number of Treg that cause this inflammatory response in the lung, they have to test this hypothesis. The manuscript is poorly written and it is difficult to follow the rationale. The data in different figures often contradict each other.

1. From Fig. 1A, it is concluded that the levels of unneddylated Cul5 decreased upon activation, while the neddylation fraction increases. However, the results in Fig. 1B do not support these conclusions since the levels of unneddylated Cul5 are the same or higher. There is inconsistency in the results among figures. This raises questions regarding the reproducibility of the data and conclusions.

To clarify, the setup of the experiments in Fig. 1a and 1b are different. Figure 1a validates our mass-spectrometry results (from our previous study published in Nature Immunology PMID:31061531) showing that neddylation of Cul5 is increased upon T cell stimulation. In Fig 1a we used T cells that were expanded in culture and then rested and restimulated with anti-CD3 and anti-CD28 for 4hr. This experiment measures how Cul5 neddylation changes when previously activated T cells are restimulated. The control in this experiment is 'resting' CD4+ T-cells.

The aim of Fig. 1b was to measure the kinetics of Cul5 expression and its neddylation in naïve T cells that were stimulated for the first time in culture. In this experiment naïve sorted CD4 T cells were stimulated with anti-CD3 and anti-CD28 for a longer time frame (0-72hr). The control here are unstimulated T-cells.

We found that in both conditions stimulated cells showed increased neddylation of Cul5.

Wording has been added to the result and legend section to ensure this is clear to the reader.

2. Since little is known about the CD4-Cul5 conditional mice, the authors need to examine the role in T cell development in these mice.

To study T cell development, we analyzed the cellular composition of the thymi of control and Cul5 deficient mice. These results are now shown in Fig. 1f, g. Our results support that Cul5 does not impact the overall frequencies of developing T cells in the thymus.

3. In Fig. 2, they use CD44 as a marker for effector cells, but this is not correct. CD44 is high in memory cells as well, and it is a marker for homeostatic proliferation. Thus, it cannot be concluded that Cul5 conditional mice have higher frequency of effector CD4 cells. In addition, in this figure (lung infiltration under physiological conditions) the authors need to examine the presence of other immune cells, not just CD4, including CD8, macrophages, eosinophils, neutrophils, etc.

The reviewer is correct. In the revised manuscript we have termed the cells CD44+ cells so as to avoid defining the cells using a term that connotes a function or developmental stage.

Based on this reviewer recommendation, we have provided data from an analysis of additional immune cells. Results are shown in panels as listed below.

Eosinophils- Fig. 2k, l

Macrophages- Supplementary Fig. 2e

Neutrophils- Supplementary Fig. 2l.

CD8 cells- Supplementary Fig. 1h.

4. In Fig. 3, they examine the allergic airway inflammation in response to exposure to HDM. Of course, it is not unexpected that the Cul5 conditional mice have increased allergic airway inflammation in response to HDM since they already have allergic airway inflammation in basal conditions. The key question is what is triggering the allergic airway inflammation in the absence of any insult. This is what is missing in this study, are CD4 cells autoreactive? are these generated during development?

In the revised version of manuscript, we have clarified the reason why 30-36 week old mice developed Th2 inflammation. The section is copied below:

“In the absence of Cul5, CD4+ T cells became overly sensitive to IL-4 which contributed to asthma pathogenesis. Increased IL-4 sensitivity predisposed CD4 T cells to becoming Th2 or, in the presence of TGF- β , Th9 cells. Th2 cells produce IL-4 creating a snowballing effect. Cul5 deficient cells are thus exposed to more IL-4, expanding Th2 and/or Th9 differentiation, and thus exacerbating inflammation and worsening pathology. This scenario most likely explains why 30–36-week-old Cul5 deficient mice developed pronounced Th2 inflammation compared to 8–10-week-old mice which showed little to no Th2 inflammation.”

We do not observe gross changes in T cell development, however a more thorough investigation of autoreactivity will be done. This is beyond the scope of this study.

5. There is more inconsistency among the results in Fig. 2 and Fig. 3. Fig. 2 shows increased frequency of CD4 cells producing IL-4, but Fig. 3 in PBS mice (should be equivalent to mice in Fig. 2) shows no difference in the production of IL-4.

It should be noted that in Fig. 3 mice were treated with PBS using intranasal injection while mice in Fig. 2 were not treated with PBS. Previous studies have shown that the presence of saline in lungs can impact lung inflammation (PMID:11734428, 16421365). Moreover, it has been shown that the presence of saline can alter immune cell recruitment and/or function (PMID:29253007). This likely explains the discrepancies noted by the reviewer.

Wording has been changed in revised manuscript to clarify this.

6. Regarding IL-9 production, again the major difference is in WT and Cul5 under physiological conditions, being higher in CD4 cells from Cul5. The authors need to show the presence of mast cells in the mice prior to exposure to HDM.

Mast cell data is now shown in Supplementary Fig. 2m.

7. While it is concluded that "Cul5 limits the frequency and number of Th9 cells and prevents lung remodeling and other pathologic features associated with asthma" the phenotype is not restricted to Th9 cells, since Th2 cytokines are also affected. The essential question is how Cul5 leads to accumulation of these cells in the absence of antigen.

Reviewer is right and we have changed the wording in the discussion section of the revised manuscript as stated above.

8. Most of the biochemistry studies in Fig. 4 were performed with D10 cells, but D10 cells are far from representing either Th2 or Th9. These studies must be repeated with primary Th2 cells and Th9 cells.

Key experiments that were performed in D10 cells were confirmed in primary murine or human CD4+ T cells (shown in Fig 4a, b, k and l). Key experiments in primary CD4 T cells were performed in presence of exogenous IL-4 which promotes Th2 and Th9 differentiation. We have modified result section to make this clearer.

To further address this reviewers comment we have added new result showing CIS levels in primary murine CD4 T cells cultured in the presence of exogenous IL-4 or with IL-4 blockade. Results are shown in Fig 4i, j.

9. In Figure 4, from mouse Th0 and Th9 cells they conclude that CIS upregulation is not mediated by IL-4, but they do not present data with CD4 cells under Th2 conditions. Furthermore, the results in human CD4 cells contradict their own conclusions since anti-IL-4 Ab reduce CIS levels. Thus, at the end, the reviewer (and most likely potential readers) has no clarity regarding the regulation of CIS in primary CD4 cells.

To address this reviewer's comment we now provide data showing CIS levels in primary CD4 T-cell cultures in the presence of IL-4 (Th2 condition) and with IL-4 blockade (Fig. 4i-j). These results are quite similar to those from D10 cells.

Regarding human cells, it is worth noting that CIS expression is increased in human CD4+T cells following activation (and in the presence of IL-4 blockade). However, expression increases further when IL-4 is added. This result, along with results from murine CD4 T-cells and D10 cells, suggest that both TCR signaling and IL-4 can drive CIS protein expression.

10. The authors propose that Cul5 deficiency predominantly favors Th9 differentiation. The data in the study do not support this conclusion. Based on the results from Fig. 5, all Th2, Th1 and Th9 genes were clearly upregulated in Cul5 deficient mice.

We may not have been sufficiently clear in our description of this result. In Fig. 5 d-f we show that Cul5 deficient T cells (cultured under Th9 conditions) have genes that are differentially expressed when compared to control cells. These differentially expressed genes are linked to a 'signature' of genes expressed in Th9 cells. We assessed the top hits from various TH cell subsets, (i.e top differentially expressed genes between WT Th1 and naive). Not surprisingly, there were a number of top hits that were similar across multiple Th subsets that likely are genes upregulated after T cell activation. In Fig. 5g we show Th9 genes unique genes (i.e. those that are only found to be upregulated in Th9 cells). Based on Fig. 5c and 5g we concluded that Cul5 deficient cells show increased expression of genes that are specific for the Th9 signature. We have modified result section to make this clearer.

11. If the mechanism for Cul5 to regulate Th2 differentiation is through Jak1 degradation, the response to many other cytokines using Jak1 should also be affected. The authors have to examine the signaling and response to more cytokines that use Jak1.

The discussion section explaining this is copied below:

"The question then becomes, how is Cul5 specific for IL-4R signalling and not broadly regulating other cytokine receptors that rely on JAK1? Like CIS, SOCS3 also inhibits JAK1 but only when JAK1 is activated by IL-6 receptor signalling. This specificity is enforced by the binding of SOCS3 to the gp130 receptor. This localisation allows SOCS3 to gain proximity to JAK1 (PMID: 12754506). A similar mechanism may be occurring in this context. In support of that we found that CIS associated not only with JAK1 but also with IL-4R α . This suggests a model in which CIS binds with pIL-4R α via its SH2 domain

and brings the CRL5 complex into proximity with pJAK1 leading to ubiquitination and degradation of pJak1. Alternatively, CIS and CRL5 might also directly target IL-4R α . These models are supported by other studies showing that CIS binds to phosphorylated receptors. In either case we believe that specificity of CRL5 is enforced by CRL5 association with the IL-4 receptor.”

12. There are several issues concerning the written manuscript. Poorly. written. For instance, it was difficult to find Figure legends and this is because they were placed before the Methods section. "IP'ed" does not exist as a word

In the revised version of the manuscript we have followed the journal guidelines for arranging manuscript sections.

Reviewer is right and we have removed “IP’ed” from manuscript.

Reviewer #3 (Remarks to the Author):

This is an interesting manuscript describing a novel role for Cul5. The authors propose that Cul5 regulates IL-4 receptor signaling and thereby controls the fate of Th cells. They show in ko and molecular interaction assays that mice lacking Cul5 have the propensity to develop Th2 and Th9 driven inflammation and pathology. The manuscript is very well written and the experiments are conclusive.

Comments:

1. Page 8, line 172: Please change "unprovoked" to "spontaneous" in the title of this chapter and throughout the manuscript.

All mice were housed in a specific pathogen free (SPF) facility. Since mice were not provoked by any agent or known pathogen, we used “unprovoked” in the manuscript. We find this terminology to be the most accurate based on our knowledge of the infection status of the mice, while bearing in mind that there are pathogens we do not currently test for. This terminology is consistent with what has been used in previously studies (PMID: 22561837, 1741708, 33264547, 2231850) evaluating asthma pathogenesis or cytokine storm.

2. Page 8, lines 183-190: The explanation and logic provided in this and the following sentences that goblet cell hyperplasia is caused by IL-4 and IL-5 is wrong. Instead, IL-13 is the major driver for goblet cell hyperplasia, however, IL-13 data are not shown here.

As suggested, IL-13 results have been included in Fig. 2e, f, i, j and in Fig. 3k, l.

3. Page 9, line 204: Please change phrasing to "...we challenged mice with house dust mite extract (HDM)...", because it is the extract made of mites with which you challenge. Figure 1F: Please change the graphs for lungs to representative ones. Fig 1G shows what is representative, however, in Fig 1F two extreme examples of what has been found in mice is depicted.

We have changed the wording as requested.

The representative flow plot in Fig. 1F (old version) has been changed (Fig. 1h) in the new version.

Figure 2. Change title from "unprovoked" to "spontaneous".

We have explained the reasoning above.

Figure 2: What is somewhat disturbing in this figure is that fact that the graphs from E-J have dissimilar numbers of animals. This is true not only for the determination of different data, e.g., percent positive cells and absolute numbers of IL-4 and IL-5 T cells and mouse immunoglobulin levels for IgE and IgG1 but also within the experiments showing IgE versus IgG1 levels. This should be really avoided.

We have added additional mice to maintain consistency in the number of mice used in each experiment. The number is now clearly indicated in the figure legend of the revised manuscript.

Figure 3C: Here one wonders, why no IL-9 data are presented. Later in the ductus of the manuscript IL-9 gets a very important place. It would be interesting to see whether or not IL-9 levels are also elevated in BAL.

We were unable to detect consistent and reproducible IL-9 in BAL from mice using this acute model. However, we did find increased numbers of mast cells, which is consistent with increased IL-9 levels.

Figure 5D-F and corresponding Results-section page 14 (lines 329-338): This part is difficult to understand. Especially the point that "fewer genes were identified for Th2, Th1, Th17 and Th0 cells". What exactly is meant by this, given the fact that, e.g., the Top Th2 hits shared many of the upregulated genes also seen in the Th9 Top hit genes?

We may not have been sufficiently clear in our description of this result. We wanted to show two things. One, that number of genes that were identified were higher in the Th9 cells compared to other subsets. Secondly, we wanted to show that Th9 cells have unique genes. We have modified the results section in an attempt to make this easier to understand.

REVIEWER COMMENTS

Reviewer #1 (Remarks to the Author):

The authors have answered all the Questions. It remains not addressed the relevance of these findings in humans. The authors have shown only one sample of Human CD4 (Fig 1c,4l,6e). If the authors have done that experiment in 4 subjects (as they write in the figure legends) then they should show all four subjects. Could then the authors report the clinical characteristics of these subjects and correlate with the results?

Reviewer #2 (Remarks to the Author):

The manuscript has been extensively revised and it is now acceptable for publication

Reviewer #3 (Remarks to the Author):

The authors have responded to all the questions raised, and, where asked for, they have made respective modifications to the ms which improved its quality considerably.

Point by point response to the reviewers.

Reviewer #1 (Remarks to the Author):

The authors have answered all the Questions. It remains not addressed the relevance of these findings in humans.

Response: The role of Cul5 in human allergy is poorly understood. While there is no published data describing SNPs in Cul5 that associate with human asthma, there is a published meeting abstract. In this abstract, it states that one of the strongest associations is in an SNP that localizes to Cul5 ([https://www.jacionline.org/article/S0091-6749\(08\)02977-1/fulltext#relatedArticles](https://www.jacionline.org/article/S0091-6749(08)02977-1/fulltext#relatedArticles)). This reference was not included in our manuscript as it lacks peer review. All published information on the role for Cul5 and Cis in human allergy is detailed in the manuscript.

The authors have shown only one sample of Human CD4 (Fig 1c,4l,6e). If the authors have done that experiment in 4 subjects (as they write in the figure legends) then they should show all four subjects.

Response: A total of 5 subjects were used for the experiments using human primary CD4 T cells shown in Figures 1c, 4l and 6e. Not all subjects samples were used in all experiments. In experiments for Figure 1, as was stated in the figure legends, 4 independent experiments were performed. For these experiments, 4 biologic replicates were used. The figure legend now clarifies that these were 4 independent experiments with a separate biologic replicate for each experiment. In figures 4 and 6, as was stated in the figure legends, three independent experiments were performed. The figure legend now clarifies that these were 3 independent experiments with a separate biologic replicate for each experiment. We have now added data from the replicates (excluding the representative shown in the main figures) in the supplemental figures.

Could then the authors report the clinical characteristics of these subjects and correlate with the results?

Response: Please note that, as stated in the manuscript, all human subjects were healthy controls. The samples were deidentified and the information provided on the samples is as follows:

Donor ID by author	Donor ID by Core	Birth year	HLA-A	HLA-B	HLA-DR	DPA1	DPB1	DQA1	DQB1	HLA-C
Donor #1	ND518	1983	A 02:01, A 03:01	B 14:02, B 39:01	DRB1 01:02, 07:01 DRB4 01:03N	DPA1 01:03	DPB1 04:01	DQA1 01:01, 02:01	DQB1 03:03, 05:01	C 07:02, C 08:02
Donor #2	ND569	1991	01:01, 31:01	07:02, 41:01	DRB1 03:01, 04:01, DRB3 03:01, DRB4 01:03	1:03	04:01, 104:01	03:03, 05:01	02:01, 03:01	07:02, 17:01
Donor #3	ND561	1994	A 23:01, 30:01	B 13:02, B 49:01	DRB1 04:03, 07:01, DRB4 01:03	DPA1 01:03, 02:01	DPB1 04:01, 17:01	DQA1 02:01, 03:01	DQB1 02:02, 03:02	C 06:02, 07:01
Donor #4	ND552	1993	A 01:01, 02:01	B 07:02, 08:01	DRB1 03:01, 11:01, DRB3 01:01, 02:02	DPA1 01:03	DPB1 02:01, 04:01	DQA1 05:01, 05:05	DQB1 02:01, 03:01	C 07:01, 07:02
Donor #5	ND570	1996	2:01	07:02, 15:01	DRB1 04:01, 15:01 DRB4 01:03, DRB5 01:01	1:03	4:01	01:02, 03:01	03:02, 06:02	03:04, 07:02

REVIEWERS' COMMENTS

Reviewer #1 (Remarks to the Author):

Revision

I appreciate to see the 3-4 healthy subjects analyzed. However, the variability of the results and the small changes observed support the notion that more control subjects should be analyzed. In addition what is missing completely is the analysis of asthmatic patients. The authors should have analyzed asthmatics and compared those results with controls in human CD4+ T cells.

Specific comments:

The authors showed the human data as:

Supplementary figure 1a-d and Fig 1c. The authors claimed in the manuscript and supplementary figure legends (marked) that they found an upregulation of Cul5 expression and neddylation following stimulation.

In those figures they showed that all 3 conditions were activated with antiCD3/CD28 for 0,24,48 hours. Do the authors mean unstimulated for 0 hours?

Also the blot shown in figure 1c was not quantified along with the replicated in supplementary Fig 1b.

The replicate in Supplementary figure 1b do not show an upregulation of the neddylation form.

I see:

Donor 1 the blot shows a little bit of induction of the neddylated form but a decrease of the unnedylated form

Donor 3: no change of the neddylated form over time

Donor 4 : a decrease of the neddylated and unnedylated form as compared to 0 hours stimulation

Also the figure legend should say unnedylated (c) and neddylated (d).

It is thus difficult to make a conclusion on the significant in humans based on these 4 subjects in which not clear results are shown.

A bigger population is required for statistical reasons.

The Supplementary figure 4 is the replicate of Figure 4l.

In figure 4l the pSTAT6 looks cut on the band, could the authors cut in a lower position and show the STAT6 blot as control?

In the supplementary data the authors have 2 antibodies shown in one blot: pSTAT6 and Cul5: how can the authors be sure that one is pSTAT6 and the other Cul5 if they do not have the recombinant or the positive control on the blot?

Also here only 3 donors were shown. The same in figure 6e and replicates Supplementary Fig 6d.

I could not see the table with the data of the patients to be correlated with the results of this manuscript.

Murine Methods: the authors say: HDM was given for 5 consecutive days 8-10? Should'nt that be 3 days? Could the authors make a drawing of their experimental asthma design in figures 2,3.

Revision

I appreciate to see the 3-4 healthy subjects analyzed. However, the variability of the results and the small changes observed support the notion that more control subjects should be analyzed. In addition what is missing completely is the analysis of asthmatic patients. The authors should have analyzed asthmatics and compared those results with controls in human CD4+ T cells.

Response: We thank reviewer for these suggestions, however, a study of how Cul5 regulates human asthma is beyond the scope of this study. Based on the concerns on variability within the data collected from cells isolated from healthy controls, we have removed these data from the manuscript.

Specific comments:

The authors showed the human data as:

Supplementary figure 1a-d and Fig 1c. The authors claimed in the manuscript and supplementary figure legends (marked) that they found an upregulation of Cul5 expression and neddylation following stimulation.

In those figures they showed that all 3 conditions were activated with antiCD3/CD28 for 0,24,48 hours. Do the authors mean unstimulated for 0 hours?

Also the blot shown in figure 1c was not quantified along with the replicated in supplementary Fig 1b.

The replicate in Supplementary figure 1b do not show an upregulation of the neddylation form. I see:

Donor 1 the blot shows a little bit of induction of the neddylated form but a decrease of the unneddylated form

Donor 3: no change of the neddylated form over time

Donor 4 : a decrease of the neddylated and unneddylated form as compared to 0 hours stimulation

Also the figure legend should say unneddylated (c) and neddylated (d).

It is thus difficult to make a conclusion on the significant in humans based on these 4 subjects in which not clear results are shown.

A bigger population is required for statistical reasons.

The Supplementary figure 4 is the replicate of Figure 4I.

In figure 4I the pSTAT6 looks cut on the band, could the authors cut in a lower position and show the STAT6 blot as control?

In the supplementary data the authors have 2 antibodies shown in one blot: pSTAT6 and Cul5: how can the authors be sure that one is pSTAT6 and the other Cul5 if they do not have the recombinant or the positive control on the blot?

Also here only 3 donors were shown. The same in figure 6e and replicates Supplementary Fig 6d.

Response: We thank reviewer for suggestions. As recommended by the editor, we have removed human CD4 T cell data from the manuscript.

I could not see the table with the data of the patients to be correlated with the results of this manuscript.

Response: In this manuscript we have used CD4+ T cells from healthy donors, not patients.

Murine Methods: the authors say: HDM was given for 5 consecutive days 8-10? Should'nt that be 3 days? Could the authors make a drawing of their experimental asthma design in figures 2,3.

Response: Mice were instilled with HDM for 5 consecutive days 8-12. In the revised manuscript we have corrected this oversight. The model for HDM has been provided in supplementary figure 3a. HDM was not used in figure 2.